# Semi-Supervised Sparse Gaussian Classification: Provable Benefits of Unlabeled Data

**Eyar Azar**
Weizmann Institute of Science
`eyar.azar@weizmann.ac.il`

**Boaz Nadler**
Weizmann Institute of Science
`boaz.nadler@weizmann.ac.il`

## Abstract

The premise of semi-supervised learning (SSL) is that combining labeled and unlabeled data yields significantly more accurate models. Despite empirical successes, the theoretical understanding of SSL is still far from complete. In this work, we study SSL for high dimensional sparse Gaussian classification. To construct an accurate classifier a key task is feature selection, detecting the few variables that separate the two classes. For this SSL setting, we analyze information theoretic lower bounds for accurate feature selection as well as computational lower bounds, assuming the low-degree likelihood hardness conjecture. Our key contribution is the identification of a regime in the problem parameters (dimension, sparsity, number of labeled and unlabeled samples) where SSL is guaranteed to be advantageous for classification. Specifically, there is a regime where it is possible to construct in polynomial time an accurate SSL classifier. However, any computationally efficient supervised or unsupervised learning schemes, that separately use only the labeled or unlabeled data would fail. Our work highlights the provable benefits of combining labeled and unlabeled data for classification and feature selection in high dimensions. We present simulations that complement our theoretical analysis.

## 1 Introduction

The presumption underlying Semi-Supervised Learning (SSL) is that more accurate predictors may be learned by leveraging both labeled and unlabeled data. Over the past 20 years, many SSL methods have been proposed and studied (Chapelle et al., 2006; Zhu & Goldberg, 2009). Indeed, on many datasets SSL yields significant improvements over supervised learning (SL) and over unsupervised learning (UL). However, there are also cases where unlabeled data does not seem to help. A fundamental theoretical issue in SSL is thus to understand under which settings can unlabeled data help to construct more accurate predictors and under which its benefit, if any, is negligible.

To address this issue, SSL was studied theoretically under various models. Several works proved that under a cluster or a manifold assumption, with sufficient unlabeled data, SSL significantly outperforms SL (Rigollet, 2007; Singh et al., 2008). In some cases, however, SSL performs similarly to UL (i.e., clustering, up to a label permutation ambiguity). In addition, Ben-David et al. (2008) described a family of distributions where SSL achieves the same error rate as SL.

In the context of the cluster assumption, a popular model for theoretical analysis is Gaussian classification, in particular binary classification for a mixture of two spherical Gaussians. In this case, the label $Y \in \{\pm 1\}$ has probabilities $\mathbb{P}(Y = y) = \pi_y$ and conditional on a label value $y$, the vector $\boldsymbol{x} \in \mathbb{R}^p$ follows a Gaussian distribution,

$$\boldsymbol{x}|y \sim \mathcal{N}(\boldsymbol{\mu}_y, \mathbf{I}_p) \tag{1}$$

where $\boldsymbol{\mu}_1, \boldsymbol{\mu}_{-1} \in \mathbb{R}^p$ are both unknown. This model and related ones were studied theoretically in supervised, unsupervised and semi-supervised settings, see for example Li et al. (2017); Tifrea et al.

38th Conference on Neural Information Processing Systems (NeurIPS 2024).

(2023); Wu & Zhou (2021), and references therein. Without assuming structure on the vectors $\boldsymbol{\mu}_y$ or on their difference (such as sparsity), there are computationally efficient SL and UL algorithms that achieve the corresponding minimax rates. Moreover, Tifrea et al. (2023) proved that for the model (1), no SSL algorithm simultaneously improves upon the minimax-optimal error rates of SL and UL. In simple words, there do not seem to be major benefits for SSL under the model (1).

In this paper we consider a mixture of two Gaussians in a *sparse* high dimensional setting. Specifically, we study balanced binary classification with a sparse difference in the class means, which is a specific instance of (1). Here, the joint distribution of a labeled sample $(\boldsymbol{x}, y)$ is given by

$$y \sim \text{Unif}\{\pm 1\}, \quad \boldsymbol{x}|y \sim \mathcal{N}(\boldsymbol{\mu}_y, \mathbf{I}_p). \tag{2}$$

The class means $\boldsymbol{\mu}_1, \boldsymbol{\mu}_{-1} \in \mathbb{R}^p$ are unknown, but their difference $\Delta\boldsymbol{\mu} = \boldsymbol{\mu}_1 - \boldsymbol{\mu}_{-1}$ is assumed to be $k$-sparse, with $k \ll p$. In a supervised setting, model (2) is closely related to the sparse normal means problem, for which both minimax rates and computationally efficient (thresholding based) algorithms have been developed and analyzed, see e.g. Johnstone (1994, 2002). In an unsupervised setting, inference on the model (2) is closely related to clustering and learning mixtures of Gaussians (Azizyan et al., 2013; Jin & Wang, 2016). A key finding is that in an unsupervised setting with a sparsity assumption, there is a statistical-computational gap (Fan et al., 2018; Löffler et al., 2022). Specifically, from an information viewpoint a number of unlabeled samples $n$ proportional to $k$ suffices to accurately cluster and to detect the support of $\Delta\boldsymbol{\mu}$. However, under under various hardness conjectures, unless $n \propto k^2$, no polynomial time algorithm is able to even detect if the data came from one or from two Gaussians (namely, distinguish between $\Delta\boldsymbol{\mu} = \boldsymbol{0}$ and $\|\Delta\boldsymbol{\mu}\| = O(1)$).

In this work we study the model (2) in a SSL setting, given $L$ labeled samples and $n$ unlabeled samples, all i.i.d. from (2). Despite extensive works on the SL and UL settings for the model (2), the corresponding SSL setting has received relatively little attention so far. This gives rise to several questions: On the theoretical front, what is the information lower bound for accurate classification and for recovering the support of $\Delta\boldsymbol{\mu}$? On the computational side, is there a computational-statistical gap in SSL? In addition, are there values of $L$ and $n$ for which SSL is provably beneficial as compared to SL and UL separately?

**Our Contributions.**    (i) We derive information theoretic lower bounds for exact support recovery in the SSL setting. As described in Section 2.2, our lower bounds characterize sets of values for the number of labeled and unlabeled samples, where any estimator based on both types of data is unable to recover the support. To derive these bounds, we view SSL as a data fusion problem involving the merging of samples that come from two measurement modalities: the labeled set and the unlabeled set. In Theorem 2.3 we present a general non-asymptotic information-theoretic result for recovering a discrete parameter in this setting. This general result is applicable to other data fusion problems and may thus be of independent interest.

(ii) We present SSL computational lower bounds. These are based on the low-degree likelihood ratio hardness conjecture (Hopkins & Steurer, 2017; Kunisky et al., 2022), in an asymptotic setting where dimension $p \to \infty$ and a suitable scaling of the sparsity $k$, and of the number of labeled and unlabeled samples. Our main result is that there is a region of the number of labeled and unlabeled samples, whereby in a SSL setting, accurate classification and feature selection are computationally hard. Our analysis extends to the SSL case previous computational lower bounds that were derived only in UL settings. In particular, if the number of the labeled samples is too small then the statistical-computational gap still remains. To the best of our knowledge, our work is the first to extend this framework to a SSL setting.

(iii) Building upon (i) and (ii), our key contribution is the identification of a region where SSL is provably computationally advantageous for classification and feature selection. Specifically, in Section 3 we develop a polynomial time SSL algorithm, denoted LSPCA, to recover the support of $\Delta\boldsymbol{\mu}$ and consequently construct a linear classifier. We then prove that in a suitable region for the number of labeled and unlabeled samples, LSPCA succeeds in both feature selection and accurate classification. In contrast, under the low degree ratio hardness conjecture, any computationally efficient SL or UL schemes, that use only the labeled or unlabeled data separately, would fail. In Section 4 we show via simulations the superiority of LSPCA, in both support recovery and classification error, in comparison to several SL and UL methods, a self-training SSL scheme and the SSL method of Zhao et al. (2008).

Figure 1 summarizes the picture emerging from our work in combination with previous papers that analyzed the UL and SL settings of (2), namely, the $x$-axis and $y$-axis in Figure 1. As in prior

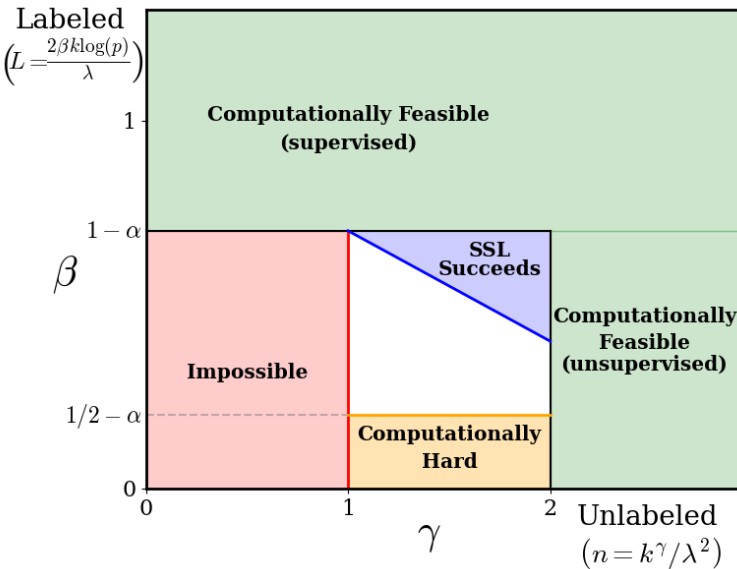

Figure 1: Semi-supervised classification and support recovery regions. The red and green regions follow from previous works. Contributions of our work include identification of the orange and the blue regions.

works we consider a fixed separation $\lambda = \|\Delta\boldsymbol{\mu}\|_2^2/4 = O(1)$, where $\Delta\boldsymbol{\mu}$ is $k$-sparse. The asymptotic setting is that $(k, L, n, p)$ all tend to infinity with the following scaling, which arises naturally for this problem (see Section 2): the number of labeled samples is $L = \lfloor 2k\beta \log(p)/\lambda \rfloor$, the number of unlabeled samples scales as $n \propto k^\gamma/\lambda^2$, and the sparsity scales as $k \propto p^\alpha$ for some $\alpha \in (0, 1/2)$. The figure shows different regions in the $(\gamma, \beta)$ plane, namely as a function of the number of unlabeled and labeled samples, where classification and feature selection are either impossible, hard or computationally easy. We say that classification is impossible if for any classifier there is a $k$-sparse vector $\Delta\boldsymbol{\mu}$ whose corresponding accuracy is no better than random. Similarly, we say that feature selection is impossible if for any estimator $\hat{S}$ of size $k$ there is a $k$-sparse $\Delta\boldsymbol{\mu}$ with support $S$ such that $|\hat{S} \cap S|/k \to 0$ as $p \to \infty$. Feature selection is easy if it is possible to construct in polynomial time a set $\hat{S}$ of size $k$ such that $|\hat{S} \cap S|/k \to 1$. This implies that the corresponding classifier has an excess risk that asymptotically tends to zero. The green region $\gamma \geq 2$ follows from Deshpande & Montanari (2014), since in this case support estimation is computationally feasible using only the unlabeled data. The region depicted in red is where classification and support recovery are impossible. The impossibility of support recovery follows from Ingster (1997); Donoho & Jin (2004), who proved that support recovery is feasible if and only if $\beta > 1 - \alpha$. The same condition holds for classification as well, as described in the supplement. The orange and blue regions in Figure 1 follow from novel results of this paper. In the orange region, defined as $\beta < 1 - \alpha$ and $1 < \gamma < 2$, our computational lower bound in Theorem 2.6 suggests that any polynomial-time scheme will not succeed in accurate classification. In the blue region, characterized by $\beta \in (1 - \gamma\alpha, 1 - \alpha)$ and $1 < \gamma < 2$, our proposed polynomial time SSL method is guaranteed to construct an accurate classifier. This is proven in Theorem 3.2. In addition, note that in this regime, the availability of unlabeled data allows to decrease the number of labeled samples by a factor of $\frac{1-\alpha}{1-\gamma\alpha}$. Under the low degree hardness conjecture, in this blue region no computationally efficient SL or UL method that separately analyze the labeled or unlabeled samples, respectively, would succeed. We conjecture that in the remaining white region, no polynomial-time algorithm exists that is able to recover the support or able to construct an accurate classifier. In summary, our work highlights the provable computational benefits of combining labeled and unlabeled data for classification and feature selection in a high dimensional sparse setting.

**Notation** For an integer $p$, we write $[p] = \{1, ..., p\}$. The cardinality of a set $B$ is $|B|$. For a vector $\boldsymbol{v} \in \mathbb{R}^p$, we denote its restriction to a subset $T \subset [p]$ by $\boldsymbol{v}|_T$. For vectors $\boldsymbol{a}, \boldsymbol{b}$, their inner product is $\langle \boldsymbol{a}, \boldsymbol{b} \rangle$, and $\|\boldsymbol{a}\|$ denotes the $\ell_2$ norm. We say that $f(p) = \omega(g(p))$ if for any $c > 0$, there exists $p_0 \geq 1$ such that $f(p) > cg(p)$ for every $p \geq p_0$.

## 2 Theoretical Results

In this section we present our first two contributions, namely an information-theoretic lower bound for exact support recovery of $\Delta\boldsymbol{\mu}$, and a computational lower bound for classification and support recovery, in a SSL setting. To this end, in Section 2.1 we first review lower bounds for SL and UL settings. As we were not able to find these precise results in the literature, for completeness we present their proofs, based on Fano's inequality, in the supplementary. Our main contribution here, described in Section 2.2, is a SSL lower bound. To derive it, we view SSL as a data fusion problem with two types of data (the labeled set and the unlabeled set). The SSL lower bound then follows by a combination of the lower bounds for SL and UL.

To derive lower bounds, it suffices to consider a specific instance of (2), where the two Gaussian means are symmetric around the origin, with $\boldsymbol{\mu}_1 = -\boldsymbol{\mu}_{-1} = \boldsymbol{\mu}$. Hence,

$$y \sim \text{Unif}\{\pm 1\}, \quad \boldsymbol{x}|y \sim \mathcal{N}(y\boldsymbol{\mu}, \mathbf{I}_p). \tag{3}$$

Here, $\boldsymbol{\mu} \in \mathbb{R}^p$ is an unknown $k$-sparse vector with $\ell_2$ norm of $\sqrt{\lambda}$. We denote its support by $S = \text{supp}(\boldsymbol{\mu}) = \{i|\mu_i \neq 0\}$, and by $\mathbb{S}$ the set of all $\binom{p}{k}$ possible $k$-sparse support sets. We denote by $\mathcal{D}_L = \{(\boldsymbol{x}_i, y_i)\}_{i=1}^{L}$ and $\mathcal{D}_n = \{\boldsymbol{x}_i\}_{i=L+1}^{L+n}$ the i.i.d. labeled and the unlabeled datasets, respectively.

To derive information and computational lower bounds for support recovery, it is necessary to impose a lower bound on $\min_{i \in S} |\mu_i|$. As in Amini & Wainwright (2009); Krauthgamer et al. (2015), it suffices to study the set of most difficult $k$-sparse vectors with such a lower bound on their entries. In our case this translates to the nonzero entries of $\boldsymbol{\mu}$ belonging to $\{\pm\sqrt{\lambda/k}\}$. Clearly, if some signal coordinates had magnitudes larger that $\sqrt{\lambda/k}$, then the problem of detecting them and constructing an accurate classifier would both be easier. Throughout our analysis, we assume $\boldsymbol{\mu}$ is of this form and the sparsity $k$ is known. All proofs appear in the supplementary.

### 2.1 Information Lower Bounds (Supervised and Unsupervised)

The next theorem states a non-asymptotic result for exact support recovery in the SL case.

**Theorem 2.1.** *Fix $\delta \in (0, 1)$. For any $(L, p, k)$ such that*

$$L < \frac{2(1-\delta)k}{\lambda} \log(p - k + 1), \tag{4}$$

*and for any support estimator $\hat{S}$ based on $\mathcal{D}_L$, it follows that $\max_{S \in \mathbb{S}} \mathbb{P}\left(\hat{S} \neq S\right) > \delta - \frac{\log 2}{\log(p-k+1)}$.*

Donoho & Jin (2004) proved a similar result in an asymptotic regime. Specifically, they proved that for $k = p^\alpha$ and $L = \frac{2\beta k}{\lambda} \log p$, approximate support recovery is possible if and only if $\beta > 1 - \alpha$. Theorem 2.1 states a stronger non-asymptotic result for exact support recovery. It implies that even if $\beta > 1 - \alpha$, it is still impossible to recover the exact support with probability tending to one if $\beta < 1$.

Next we present an information lower bound for exact support recovery in UL. Here we observe $n$ vectors $\boldsymbol{x}_i$ from (3) but not their labels $y_i$.

**Theorem 2.2.** *Fix $\delta \in (0, 1)$. For any $(n, p, k) \to \infty$ with $\frac{k}{p} \to 0$ and*

$$n < \frac{2(1-\delta)k}{\lambda^2} \log(p - k + 1) \max\{1, \lambda\}, \tag{5}$$

*for any support estimator $\hat{S}$ based on $\mathcal{D}_n$, as $p \to \infty$, then $\max_{S \in \mathbb{S}} \mathbb{P}\left(\hat{S} \neq S\right) \geq \delta$.*

The scaling in Eq. (5) appeared in several prior works on related problems. Azizyan et al. (2013) showed that for $\lambda < 1$, with number of samples $n < C\frac{k}{\lambda^2} \log(p/k)$, no clustering method can achieve accuracy better than random. Verzelen & Arias-Castro (2017) studied hypothesis testing whether the data came from a single Gaussian or from a mixture of two Gaussians. In Proposition 3 of their paper, they proved that for $n \leq \frac{k}{\lambda^2} \log(\frac{ep}{k}) \max\{1, \lambda\}$, any testing procedure is asymptotically powerless. Note that for $k = p^\alpha$ with $\alpha < 1$, the lower bound derived in Verzelen & Arias-Castro (2017) has a similar form to (5) with a factor of $1 - \alpha$, which is slightly smaller. Thus the bound in (5) is sharper.

## 2.2 Semi-Supervised Setting

In the SSL case, the observed data consists of two subsets, one with $L$ labeled samples and the other with $n$ unlabeled ones. We now develop information-theoretic and computational lower bounds for this setting. The information lower bound is based on the results in Section 2.1 for SL and UL settings. The computational lower bound relies on the low-degree likelihood hardness conjecture. Over the past 10 years, several authors studied statistical-computational gaps for various high dimensional problems. For the sparse Gaussian mixture (3) both Fan et al. (2018) and Löffler et al. (2022) derived such gaps in an UL setting. To the best of our knowledge, our work is amongst the first to explore computational-statistical gaps in a SSL setting. Our analysis, described below, shows that with relatively few labeled samples, the computational statistical gap continues to hold. In contrast, as we describe in Section 3, with a sufficiently large number of labeled samples, but not enough so solve the problem using only the labeled set, the computational-statistical gap is resolved. In particular, we present a polynomial time SSL algorithm that bridges this gap.

**Information Lower Bounds.** Before presenting results for the mixture model (3), we analyze a more general case. We study the recovery of a latent variable $S$ that belongs to a large finite set $\mathbb{S}$, given measurements from two different modalities. Formally, the problem is to recover $S$ from two independent sets of samples $\{x_i\}_{i=1}^N$ and $\{z_j\}_{j=1}^M$ of the following form,

$$\{x_i\}_{i=1}^N \sim f_x(x|S), \quad \{z_j\}_{j=1}^M \sim g_z(z|S). \tag{6}$$

Here, $f_x(x|S)$ and $g_z(z|S)$ are known probability density functions. These functions encode information on $S$ from the two types of measurements. In our SSL setting, $x$ represents an unlabeled sample, whereas $z = (x, y)$ a labeled one, and $S$ is the unknown support of $\mu$.

Our goal is to derive information lower bounds for this problem. To this end, we assume that $S$ is a random variable uniformly distributed over a finite set $\mathbb{S}$, and denote by $I_x = I(x; S)$ and $I_z = I(z; S)$ the mutual information of $x$ with $S$ and of $z$ with $S$, respectively. Further, recall a classical result in information theory that to recover $S$ from $N$ i.i.d. samples of $x$, $N$ must scale at least linearly with $\frac{\log |\mathbb{S}|}{I_x}$. A similar argument applies to $z$. For further details, see Scarlett & Cevher (2021). The following theorem states a general non-asymptotic information-theoretic result for recovering $S$ from the above two sets of samples. Hence, it is applicable to other problems involving data fusion from multiple sources and may thus be of independent interest.

**Theorem 2.3.** *Fix $\delta \in (0, 1)$. Let $N, M$ be integers that satisfy $\max\{N \cdot I_x, M \cdot I_z\} < (1 - \delta) \log |\mathbb{S}|$. Let $N_q = \lfloor qN \rfloor$ and $M_q = \lfloor (1 - q)M \rfloor$, for $q \in [0, 1]$. Then, any estimator $\hat{S}$ based on $\{x_i\}_{i=1}^{N_q}$ and $\{z_j\}_{j=1}^{M_q}$ satisfies*

$$\mathbb{P}\left(\hat{S} \neq S\right) > \delta - \frac{\log 2}{\log |\mathbb{S}|}.$$

This theorem implies that with any convex combination of samples from the two modalities, $qN$ from the first and $(1 - q)M$ from the second, accurate recovery of $S$ is not possible if $N$ and $M$ are both too small. Essentially this follows from the additivity of mutual information.

Combining Theorem 2.3 with the proofs of Theorems 2.1 and 2.2 yields the following information lower bound for the semi-supervised case.

**Corollary 2.4.** *Let $\mathcal{D}_L$ and $\mathcal{D}_n$ be sets of $L$ and $n$ i.i.d. labeled and unlabeled samples from the mixture model (3). Fix $\delta \in (0, 1)$. Let $(L_0, n_0, p, k) \to \infty$, with $\frac{k}{p} \to 0$, be such that*

$$L_0 < \frac{2(1 - \delta)k}{\lambda} \log(p - k + 1), \quad n_0 < \frac{2(1 - \delta)k}{\lambda^2} \log(p - k + 1) \max\{1, \lambda\}.$$

*Suppose the number of labeled and unlabeled samples satisfy $L = \lfloor qL_0 \rfloor$ and $n = \lfloor (1 - q)n_0 \rfloor$ for some $q \in [0, 1]$. Then, for any estimator $\hat{S}$ based on $\mathcal{D}_L \cup \mathcal{D}_n$, as $p \to \infty$*

$$\max_{S \in \mathbb{S}} \mathbb{P}\left(\hat{S} \neq S\right) \geq \delta.$$

**Computational Lower Bounds.** Our SSL computational lower bound is based on the low-degree framework, and its associated hardness conjecture (Hopkins & Steurer, 2017; Kunisky et al., 2022). This framework was used to derive computational lower bounds for various unsupervised high dimensional problems including sparse-PCA and sparse Gaussian mixture models (Löffler et al., 2022; Schramm & Wein, 2022; Ding et al., 2023). To the best of our knowledge, our work is the first to adapt this framework to a SSL setting.

For our paper to be self-contained, we first briefly describe this framework and its hardness conjecture. We then present its adaptation to our SSL setting. The low degree likelihood framework focuses on unsupervised *detection* problems, specifically the ability to distinguish between two distributions $\mathbb{P}$ and $\mathbb{Q}$, given $n$ i.i.d. samples. Specifically, denote the null distribution of $n$ samples by $\mathbb{Q}_n$, whereby all $\boldsymbol{x}_i \sim \mathbb{Q}$, and denote by $\mathbb{P}_n$ the alternative distribution, with all $\boldsymbol{x}_i \sim \mathbb{P}$.

Under the low-degree framework, one analyzes how well can the distributions $\mathbb{P}_n$ and $\mathbb{Q}_n$ be distinguished by a low-degree multivariate polynomial $f : \mathbb{R}^{p \times n} \to \mathbb{R}$. The idea is to construct a polynomial $f$ which attains large values for data from $\mathbb{P}_n$ and small values for data from $\mathbb{Q}_n$. Specifically, the following metric plays a key role in this framework,

$$\|\mathcal{L}_n^{\leq D}\| := \max_{\deg(f) \leq D} \frac{\mathbb{E}_{X \sim \mathbb{P}_n}[f(X)]}{\sqrt{\mathbb{E}_{X \sim \mathbb{Q}_n}[f(X)^2]}}, \tag{7}$$

where the maximum is over polynomials $f$ of degree at most $D$. The value $\|\mathcal{L}_n^{\leq D}\|$ characterizes how well degree-$D$ polynomials can distinguish $\mathbb{P}_n$ from $\mathbb{Q}_n$. If $\|\mathcal{L}_n^{\leq D}\| = O(1)$, then $\mathbb{P}_n$ and $\mathbb{Q}_n$ cannot be distinguished via a degree-$D$ polynomial. Computational hardness results that use the low-degree framework are based on the following conjecture, which we here state informally, and refer the reader to Löffler et al. (2022) for its precise statement.

*Conjecture* 2.5 (Informal). Let $\mathbb{Q}_n$ and $\mathbb{P}_n$ be two distinct distributions. Suppose that there exists $D = \omega(\log(pn))$ for which $\|\mathcal{L}_n^{\leq D}\|$ remains bounded as $p \to \infty$. Then, there is no polynomial-time test $T : \mathbb{R}^{p \times n} \to \{0, 1\}$ that satisfies

$$\mathbb{E}_{X \sim \mathbb{P}_n}[T(X)] + \mathbb{E}_{X \sim \mathbb{Q}_n}[1 - T(X)] = o(1).$$

In simple words, Conjecture 2.5 states that if $\|\mathcal{L}_n^{\leq D}\| = O(1)$ as $p \to \infty$, then it is not possible to distinguish between $\mathbb{P}_n$ and $\mathbb{Q}_n$ using a polynomial-time algorithm, as no test has both a low false alarm as well as a low mis-detection rate (the two terms in the equation above).

We now show how to extend this framework, focused on unsupervised detection, to our SSL setting. To this end, consider $L + n$ samples, distributed according to either a null distribution $\mathbb{Q}_{L+n}$ or an alternative distribution $\mathbb{P}_{L+n}$. In our case, the null distribution is

$$\mathbb{Q}_{L+n}: \quad \boldsymbol{x}_i = \boldsymbol{\xi}_i \sim \mathcal{N}(0, \mathbf{I}_p), \ i \in [L+n], \tag{8}$$

whereas the alternative belongs to the following set of distributions,

$$\mathbb{P}_{L+n}: \quad \begin{cases} \boldsymbol{x}_i = \boldsymbol{\mu}^S + \boldsymbol{\xi}_i, & i \in [L], \\ \boldsymbol{x}_i = y_i \boldsymbol{\mu}^S + \boldsymbol{\xi}_i, & L < i \leq L+n. \end{cases} \tag{9}$$

Here, $S$ is uniformly distributed over $\mathbb{S}$, $\boldsymbol{\mu}^S(j) = \sqrt{\frac{\lambda}{k}} \mathbb{1}\{j \in S\}$, and $y_i$ are unobserved Rademacher random variables.

The next theorem presents a low-degree bound for our SSL testing problem. The scalings of $L, n$ and $k$ with $p$ and $\lambda$ are motivated by those appearing in Theorems 2.1 and 2.2.

**Theorem 2.6.** *Let* $k = \lfloor c_1 p^\alpha \rfloor$, $L = \lfloor \frac{2\beta k}{\lambda} \log(p - k) \rfloor$, $n = \lfloor c_2 \frac{k^\gamma}{\lambda^2} \rfloor$ *and* $D = (\log p)^2$, *for some* $\beta, \gamma, \lambda, c_1, c_2 \in \mathbb{R}_+$ *and* $\alpha \in (0, \frac{1}{2})$. *With the null and alternative distributions defined in* (8) *and* (9), *if* $\beta < \frac{1}{2} - \alpha$ *and* $\gamma < 2$, *then as* $p \to \infty$

$$\|\mathcal{L}_{L+n}^{\leq D}\|^2 = O(1).$$

Theorem 2.6 together with the hardness conjecture 2.5 extends to the SSL case previous computational lower bounds that were derived only in UL settings ($\beta = 0$) (Fan et al., 2018; Löffler et al., 2022). Next, we make several remarks regarding the theorem.

**SSL statistical-computational gap.** In the rectangular region $\beta < \frac{1}{2} - \alpha$ and $1 < \gamma < 2$, depicted in orange in Figure 1, under the hardness conjecture 2.5, distinguishing between $\mathbb{P}$ and $\mathbb{Q}$ is computationally hard. Since testing is easier than variable selection and classification (Verzelen & Arias-Castro, 2017; Fan et al., 2018), in this region these tasks are computationally hard as well.

**Tightness of condition $\gamma < 2$ in Theorem 2.6.** This condition is sharp, since for $\gamma \geq 2$, namely $n \gtrsim \frac{k^2}{\lambda^2}$, the support can be recovered by a polynomial-time algorithm, such as thresholding the covariance matrix followed by PCA, see Deshpande & Montanari (2014) and Krauthgamer et al. (2015).

**Tightness of condition $\beta < \frac{1}{2} - \alpha$.** This condition is tight for detection, though not necessarily for feature selection or classification. The reason is that for $\beta > \frac{1}{2} - \alpha$, it is possible to distinguish between $\mathbb{P}$ and $\mathbb{Q}$, using only the labeled data (Ingster, 1997; Donoho & Jin, 2004).

Combining Theorems 2.1-2.6 leaves a rectangular region $1 < \gamma < 2$ and $\frac{1}{2} - \alpha < \beta < 1 - \alpha$ where SSL support recovery is feasible from an information view, but we do not know if it possible in a computationally efficient manner. In the next section we present a polynomial time SSL method that in part of this rectangle, depicted in blue in Figure 1, is guaranteed to recover $S$ and construct an accurate classifier. We conclude with the following conjecture regarding the remaining white region:

*Conjecture* 2.7. Let $\mathcal{D}_L, \mathcal{D}_n$ be sets of $L$ and $n$ i.i.d. labeled and unlabeled samples from the model (3). Assume, as in Theorem 2.6 that $k \propto p^\alpha$, $L = \lfloor \frac{2\beta k}{\lambda} \log(p - k) \rfloor$ and $n \propto \frac{k^\gamma}{\lambda^2}$. Then in the white region depicted in Figure 1, no polynomial-time algorithm is able to recover the support $S$ or to construct an accurate classifier.

## 3  Semi-Supervised Learning Scheme

We present a SSL scheme, denoted `LSPCA`, for the model (2), that is simple and has polynomial runtime. In subsection 3.2 we prove that in the blue region of Figure 1 it recovers the support, and thus constructs an accurate classifier. In this region, under the hardness conjecture 2.5, computationally efficient algorithms that rely solely on either labeled or unlabeled data would fail.

**Preliminaries.** To motivate the derivation of `LSPCA`, we first briefly review some properties of the sparse model (2). First, note that the covariance matrix of $\boldsymbol{x}$ is $\Sigma_x = \frac{1}{4}\Delta\boldsymbol{\mu}\Delta\boldsymbol{\mu}^\top + \mathbf{I}_p$. This is a rank-one spiked covariance model, whose leading eigenvector is $\Delta\boldsymbol{\mu}$, up to a $\pm$ sign. Hence, with enough unlabeled data, $\Delta\boldsymbol{\mu}$ can be estimated by vanilla PCA on the sample covariance or by some sparse-PCA procedure taking advantage of the sparsity of $\Delta\boldsymbol{\mu}$. Unfortunately, in high dimensions with a limited number of samples, these procedures may output quite inaccurate estimates, see for example Nadler (2008); Birnbaum et al. (2013). The main idea of our approach is to run these procedures after an initial variable screening step that uses the labeled data to reduce the dimension.

### 3.1  The `LSPCA` Scheme

Our SSL scheme, denoted `LSPCA`, stands for Label Screening PCA. As described in Algorithm 1, `LSPCA` has two input parameters: the sparsity $k$ and a variable screening factor $\tilde{\beta} < 1$. The scheme consists of two main steps: (i) removal of noise variables using the labeled samples; (ii) support estimation from the remaining variables using the unlabeled samples via PCA. Finally, a linear classifier is constructed via the leading eigenvector of the covariance matrix on the estimated support.

The first stage screens variables using only the labeled samples. While our setting is different, this stage is similar in spirit to Sure Independence Screening (SIS), which was developed for high-dimensional regression (Fan & Lv, 2008). To this end, our scheme first constructs the vector,

$$\boldsymbol{w}_L = \frac{1}{L_+} \sum_{i:y_i=1} \boldsymbol{x}_i - \frac{1}{L_-} \sum_{i:y_i=-1} \boldsymbol{x}_i, \tag{10}$$

where $L_+ = |\{i \in [L] : y_i = 1\}|$ and $L_- = L - L_+$. With a balanced mixture $\mathbb{P}(Y = \pm 1) = \frac{1}{2}$, it follows that $\boldsymbol{w}_L \approx \Delta\boldsymbol{\mu} + \frac{2}{\sqrt{L}} N(\mathbf{0}, \mathbf{I}_p)$. Hence, $\boldsymbol{w}_L$ can be viewed as a noisy estimate of $\Delta\boldsymbol{\mu}$. If the number of labeled samples were large enough, then the top $k$ coordinates of $\boldsymbol{w}_L$ would coincide with the support of $\Delta\boldsymbol{\mu}$. With few labeled samples, while not necessarily the top-$k$, the entries of $\boldsymbol{w}_L$ at

---

**Algorithm 1** LSPCA

---

**Input:** $\mathcal{D}_L = \{(\boldsymbol{x}_i, y_i)\}_{i=1}^L$, $\mathcal{D}_n = \{\boldsymbol{x}_i\}_{i=L+1}^{L+n}$, parameters $k, \tilde{\beta}$
**Step I: Labeled Data**
Compute $\boldsymbol{w}_L$ via Eq. (10)
Select its top $p^{(1-\tilde{\beta})}$ entries, denoted by $S_L$
**Step II: Unlabeled Data**
Compute the sample covariance $\hat{\Sigma}|_{S_L}$ by Eq. (11)
Compute leading eigenvector $\hat{\mathbf{v}}_{\text{PCA}}$ of $\hat{\Sigma}|_{S_L}$
Set $\hat{S}$ to be the top $k$ entries of $|\hat{\mathbf{v}}_{\text{PCA}}|$
Compute leading eigenvector $\hat{\mathbf{v}}$ of $\hat{\Sigma}|_{\hat{S}}$
**Output:** Estimated support $\hat{S}$, and vector $\hat{\mathbf{v}}$.

---

the support indices still have relative large magnitudes. Given the input parameter $\tilde{\beta} > 0$, the scheme retains the indices that correspond to the largest $p^{1-\tilde{\beta}}$ entries in absolute value of $\boldsymbol{w}_L$. We denote this set by $S_L$. Note that for any $\tilde{\beta} > 0$, this step significantly reduces the dimension (as $\tilde{\beta} > 0$ then $p^{1-\tilde{\beta}} \ll p$). In addition, as analyzed theoretically in the next section, for some parameter regimes, this step retains in $S_L$ (nearly all of) the $k$ support indices. These two properties are essential for the success of the second stage, which we now describe.

The second step estimates the support $S$ using the unlabeled data. Specifically, LSPCA constructs the sample covariance matrix restricted to the set $S_L$,

$$\hat{\Sigma}|_{S_L} = \frac{1}{n} \sum_{i=L+1}^{n+L} (\boldsymbol{x}_i - \bar{\boldsymbol{x}})|_{S_L} (\boldsymbol{x}_i - \bar{\boldsymbol{x}})|_{S_L}^\top, \tag{11}$$

where $\bar{\boldsymbol{x}} = \frac{1}{n} \sum_{i=L+1}^{n+L} \boldsymbol{x}_i$ is the empirical mean of the unlabeled data. Next, it computes the leading eigenvector $\hat{\mathbf{v}}_{\text{PCA}}$ of $\hat{\Sigma}|_{S_L}$. The output support set $\hat{S}$ consists of the $k$ indices of $\hat{\mathbf{v}}_{\text{PCA}}$ with largest magnitude. Finally, the vector $\Delta\boldsymbol{\mu}$ is (up to scaling) estimated by the leading eigenvector of $\hat{\Sigma}$ restricted to $\hat{S}$, with its sign determined by the labeled data.

*Remark* 3.1. After the removal of variables in the first step, the input dimension to the second step is much lower, $\tilde{p} = p^{1-\tilde{\beta}}$. Despite this reduction in dimension, as long as the vector $\Delta\boldsymbol{\mu}$ is sufficiently sparse with $k \ll \tilde{p}$, or equivalently $\alpha < 1 - \tilde{\beta}$, then in the second step our goal is still to find a sparse eigenvector. Hence, instead of vanilla PCA, we may replace the second step by any suitable (polynomial time) sparse-PCA procedure. We refer to this approach as LS$^2$PCA (Labeled Screening Sparse-PCA). As illustrated in the simulations, for finite sample sizes, this can lead to improved support recovery and lower classification errors.

### 3.2 Support Recovery and Classification Guarantees for LSPCA

Before presenting our main result, we first recall two standard evaluation metrics. The classification error of a classifier $C : \mathbb{R}^p \to \{\pm 1\}$ is defined as $\mathcal{R}(C) = \mathbb{P}(C(\boldsymbol{x}) \neq y)$. Its excess risk is defined as

$$\mathcal{E}(C) = \mathcal{R}(C) - \mathcal{R}^* = \mathcal{R}(C) - \inf_{C'} \mathcal{R}(C'). \tag{12}$$

As in Verzelen & Arias-Castro (2017), the accuracy of a support estimate $\hat{S}$ is defined by its normalized overlap with the true support, namely $|\hat{S} \cap S|/k$. To simplify the analysis, we focus on the symmetric setting where $\boldsymbol{\mu}_1 = -\boldsymbol{\mu}_{-1} = \boldsymbol{\mu}$. The next theorem presents theoretical guarantees for LSPCA, in terms of support recovery and the excess risk of the corresponding classifier.

**Theorem 3.2.** *Let $\mathcal{D}_L, \mathcal{D}_n$ be labeled and unlabeled sets of $L$ and $n$ i.i.d. samples from (3) with a $k$-sparse $\boldsymbol{\mu}$ whose non-zero entries are $\pm\sqrt{\lambda/k}$. Suppose that $k = \lfloor c_1 p^\alpha \rfloor, L = \lfloor \frac{2\beta k}{\lambda} \log(p-k) \rfloor, n = \lfloor c_2 \frac{k^\gamma}{\lambda^2} \rfloor$, for some fixed $0 < \alpha < 1/2, 0 < \beta < 1 - \alpha, \gamma > 1$ and $\lambda, c_1, c_2 \in \mathbb{R}_+$. Let $\hat{S}, \hat{\boldsymbol{v}}$ be the output of Algorithm 1 with input $k$ and screening factor $\tilde{\beta}$. If $\beta > 1 - \gamma\alpha$ and $\tilde{\beta} \in (1 - \gamma\alpha, \beta)$, then*

$$\lim_{p \to \infty} \mathbb{P}\left(\frac{|S \cap \hat{S}|}{k} \geq 1 - \epsilon\right) = 1, \quad \forall \epsilon > 0, \tag{13}$$

*and the excess risk of the corresponding classifier $C(\boldsymbol{x}) = \text{sign}\langle \hat{\mathbf{v}}, \boldsymbol{x}\rangle$, satisfies*

$$\lim_{p\to\infty} \mathcal{E}(C) = 0. \tag{14}$$

The interesting region where Theorem 3.2 provides a non-trivial recovery guarantee is the triangle depicted in blue in Figure 1. Indeed, in this region, LSPCA recovers the support and constructs an accurate classifier. In contrast, any SL algorithm would fail, and under the low degree hardness conjecture, any computationally efficient UL scheme would fail as well. To the best of our knowledge, our work is the first to rigorously prove the computational benefits of SSL, in bridging the computational-statistical gap in high dimensions. As mentioned above, we conjecture that in the remaining white region in Figure 1, it is not possible to construct in polynomial time an accurate SSL classifier. The intuition underlying this conjecture is based on the work of Donoho & Jin (2004), where in the fully supervised (SL) setting, the authors show that there is a detection-recovery gap. Namely for a range of number of labeled samples, it is possible to detect that a sparse signal is present, but it is not possible to reliably recover its support. Intuitively, adding a few unlabeled samples should not resolve this gap.

## 4    Simulation Results

We illustrate via several simulations some of our theoretical findings. Specifically, we compare LSPCA and LS$^2$PCA to various SL, UL and SSL schemes, in terms of both accuracy of support recovery and classification error. The sparse PCA method used in LS$^2$PCA was iteratively proxy update (IPU) (Tian et al., 2020) . We generate $L + n$ labeled and unlabeled samples according to the model (3) with a $k$-sparse $\boldsymbol{\mu}$ whose non-zero entries are $\pm\sqrt{\lambda/k}$. The quality of a support estimate $\hat{S}$ is measured by its normalized accuracy $|\hat{S} \cap S|/k$. For all methods compared we assume the sparsity $k$ is known. Hence, each method outputs a $k-$sparse unit norm vector $\hat{\boldsymbol{\mu}}$, so its corresponding linear classifier is $\boldsymbol{x} \mapsto \text{sign}\langle \hat{\boldsymbol{\mu}}, \boldsymbol{x}\rangle$. Given the model (3), its generalization error is $\Phi^c(\langle \hat{\boldsymbol{\mu}}, \boldsymbol{\mu}\rangle)$. We run our SSL schemes with $\tilde{\beta} = \beta - (\beta - (1-\gamma\alpha))/4$ which satisfies the requirements of Theorem 3.2. We present experiments with $p = 10^5, k = p^{0.4} = 100$ and $\lambda = 3$, though the behavior is similar for other settings as well. The error of the Bayes classifier is $\Phi^c(\sqrt{\lambda}) \approx 0.042$. We report the average (with $\pm 1$ standard deviation) of the support recovery accuracy and the classification error over $M = 50$ random realizations. All experiments were run on a Intel i7 CPU 2.10 GHz.

We empirically evaluate the benefit of $L = 200$ labeled samples in addition to $n$ unlabeled ones. We compare our SSL schemes LSPCA and LS$^2$PCA to the following UL methods, taking all $L + n$ samples as unlabeled: ASPCA (Birnbaum et al., 2013), and IPU (Tian et al., 2020). The SSL methods that we compare are LSDF (Zhao et al., 2008), and self-training (self-train). The self-training algorithm is similar to the approach in Oymak & Gulcu (2021), but explicitly accounts for the known sparsity $k$: (i) compute $\boldsymbol{w}_L$ of (10) using the labeled samples, and keep its $k$ largest entries, denote the result by $\boldsymbol{w}_L^{(k)}$; (ii) compute the dot products $c_i = \langle \boldsymbol{w}_L^{(k)}, \boldsymbol{x}_i\rangle$ and the pseudo labels $\tilde{y}_i = \text{sign}(c_i)$; (iii) let $n_{\text{eff}}$ be the cardinality of the set $\{i : |c_i| > \Gamma\}$, for some threshold value $\Gamma \geq 0$; (iv) estimate the support by the top-$k$ coordinates in absolute value of the following vector:

$$\boldsymbol{w}_{\text{self}} = \frac{1}{L + n_{\text{eff}}} \left( \sum_{i=1}^{L} y_i \boldsymbol{x}_i + \sum_{i=L+1}^{L+n} \mathbb{1}\{|c_i| > \Gamma\}\tilde{y}_i \boldsymbol{x}_i \right)$$

In the experiments we used $\Gamma = 0.8$, which gave the best performance. Also, we implemented the SL scheme Top-K Labeled that selects the indices of the top-$k$ entries of $|\boldsymbol{w}_L|$ of (10). As shown in the supplementary, this is the maximum-likelihood estimator for $S$ based on the labeled data.

Figure 2 illustrates our key theoretical result - that in certain cases SSL can yield accurate classification and feature selection where SL and UL simultaneously fail. The left panel of Figure 2 shows the average accuracies of support estimation for the different schemes as a function of number of unlabeled samples $n$. Except at small values of $n$,  LS$^2$PCA achieved the best accuracy out of all methods compared. The right panel shows the classification errors of the different methods. The black horizontal line is the error of the Bayes optimal (Oracle) classifier. As seen in the figure, our SSL schemes come close to the Bayes error while SL and UL schemes have much higher errors.

We present further experiments that empirically illustrate the benefit of using a fixed number of $n = 1000$ unlabeled samples while varying the number of labeled samples $L$. Specifically, we

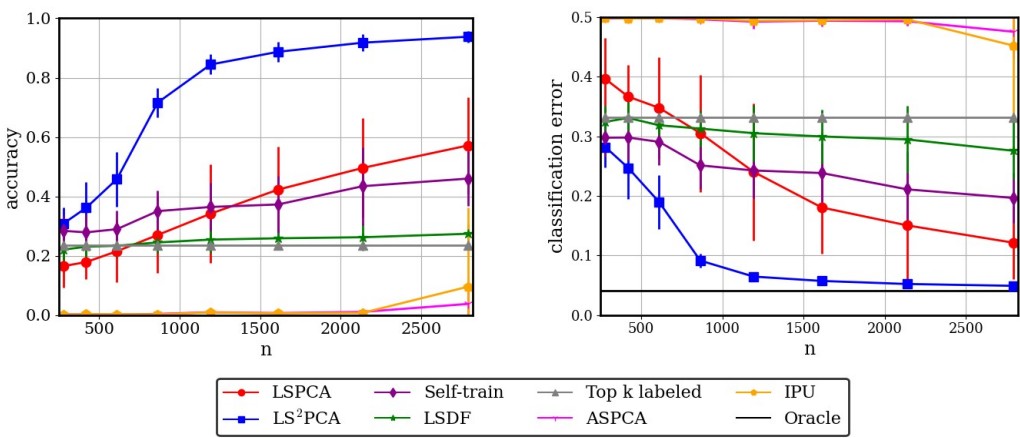

Figure 2: Empirical simulation results. (Left) Support recovery, (Right) Classification error.

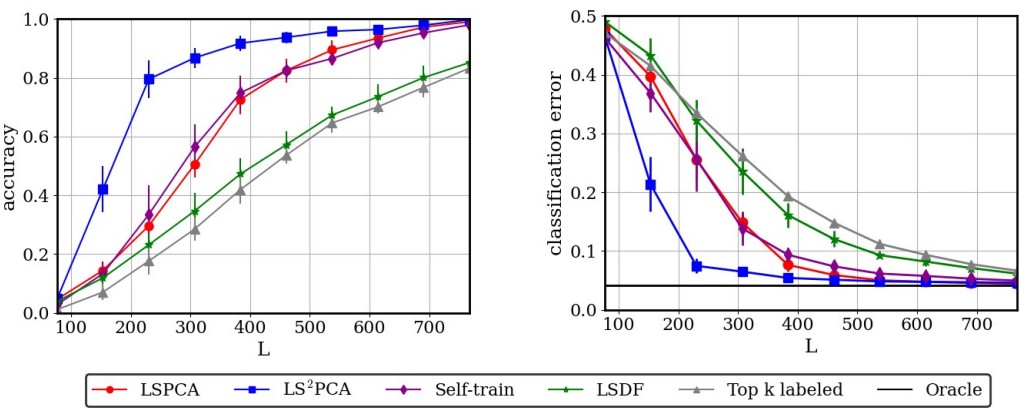

Figure 3: Empirical simulation results. (Left) Support recovery, (Right) Classification error.

compare our SSL algorithms LSPCA and LS$^2$PCA to the SSL methods self-train and LSDF, as well as to the SL scheme Top-K Labeled, which uses only the $L$ labeled samples. Figure 3 illustrates the support recovery accuracies and the classification error as a function of the number of labeled samples $L$. As seen in the figure, adding $n = 1000$ unlabeled samples significantly improves the classification and support recovery accuracies.

## 5  Summary and Discussion

In this work, we analyzed classification of a mixture of two Gaussians in a sparse high dimensional setting. Our analysis highlighted provable computational benefits of SSL. Two notable limitations of our work are that we studied a mixture of only two components, both of which are spherical Gaussians. It is thus of interest to extend our analysis to more components and to other distributions.

From a broader perspective, many SSL methods for feature selection have been proposed and shown empirically to be beneficial, see for example the review by (Sheikhpour et al., 2017). An interesting open problem is to theoretically prove their benefits, over purely SL or UL. In particular it would be interesting to find cases where SSL improves over both SL and UL in its error rates, not only computationally.

## Acknowledgements

The research of B.N. was partially supported by ISF grant 2362/22. B.N. is an incumbent of the William Petschek professorial chair of mathematics. We thank Gil Kur and Jonathan Weed for interesting discussions. We also thank the anonymous reviewers for their valuable feedback that improved the quality of our manuscript.

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

# A Auxiliary Lemmas

We first present several auxiliary lemmas used to prove our theorems. We denote the complement of the standard normal cumulative distribution function by $\Phi^c(t) = \mathbb{P}(Z > t)$, where $Z \sim \mathcal{N}(0,1)$. The following lemma states a well known upper bound on $\Phi^c$.

**Lemma A.1.** *For any $t > 1$,*

$$\Phi^c(t) \leq \frac{1}{\sqrt{2\pi}t}e^{-t^2/2}. \tag{15}$$

**Lemma A.2** (Chernoff (1952)). *Suppose $\{x_i\}_{i=1}^n$ are i.i.d. Bernoulli random variables, with $\Pr[x_i = 1] = q$. Let $X$ denote their sum. Then, for any $\delta \geq 0$,*

$$\mathbb{P}(X \geq (1+\delta)nq) \leq e^{-\frac{\delta^2 nq}{2+\delta}}, \tag{16}$$

*and for any $\delta \in [0,1]$*

$$\mathbb{P}(X \leq (1-\delta)nq) \leq e^{-\frac{\delta^2 nq}{2}}. \tag{17}$$

A common approach to prove lower bounds is using Fano's inequality. Here, we use the following version of Fano's lemma, see Yang & Barron (1999).

**Lemma A.3.** *Let $\theta$ be a random variable uniformly distributed over a finite set $\Theta$. Let $z_1, z_2, \ldots, z_n$ be $n$ i.i.d. samples from a density $f(z|\theta)$. Then, for any estimator $\hat{\theta}(z_1, \ldots, z_n) \in \Theta$,*

$$\mathbb{P}\left(\hat{\theta} \neq \theta\right) \geq 1 - \frac{I(\theta; Z^n) + \log 2}{\log |\Theta|}, \tag{18}$$

*where $I(\theta; Z^n)$ is the mutual information between $\theta$ and the samples $Z^n = (z_1, z_2, \ldots, z_n)$.*

In our proofs we use several well known properties of the entropy function. For convenience we here state some of them. First, we recall the explicit expression for the entropy of a multivariate Gaussian.

**Lemma A.4.** *Let $x \sim \mathcal{N}(\mu, \Sigma)$. Then, its entropy is given by*

$$H(x) = \frac{p}{2}(1 + \log(2\pi)) + \frac{1}{2}\log det(\Sigma). \tag{19}$$

Next, the following lemma states that the multivariate Gaussian distribution maximizes the entropy over all continuous distributions with the same covariance (Cover & Thomas, 2006, pg. 254).

**Lemma A.5.** *Let $x$ be a continuous random variable with mean $\mu \in \mathbb{R}^p$ and covariance $\Sigma \in \mathbb{R}^{p \times p}$, and let $y \sim \mathcal{N}(\mu, \Sigma)$. If the support of $x$ is all of $\mathbb{R}^p$ then*

$$H(x) \leq H(y). \tag{20}$$

The next lemma states the sub-additive property of the entropy function.

**Lemma A.6.** *Let $x$ and $y$ be jointly distributed random variables. Then,*

$$H(x, y) \leq H(x) + H(y). \tag{21}$$

To prove Theorem 2.2 we use the following lemma.

**Lemma A.7.** *Let $\lambda \in (0, 1)$, and let $k$ be a positive integer. Then, for $w \sim N(0, \frac{k-1}{k})$*

$$\mathbb{E}\left[\tanh(\lambda + \sqrt{\lambda}w) - \frac{1}{2}\tanh^2\left(\lambda + \sqrt{\lambda}w\right)\right] \leq \frac{1}{2}\left(\lambda + 3\sqrt{\lambda/k}\right) \tag{22}$$

*Proof of Lemma A.7.* Let $q(w) = \tanh(\lambda + \sqrt{\lambda}w) - \tanh^2(\lambda + \sqrt{\lambda}w)$. In terms of $q(w)$, the left hand side of (22) may be written as follows

$$\mathbb{E}\left[\tanh(\lambda + \sqrt{\lambda}w) - \frac{1}{2}\tanh^2\left(\lambda + \sqrt{\lambda}w\right)\right] = \frac{1}{2}\left(\mathbb{E}\left[\tanh(\lambda + \sqrt{\lambda}w)\right] + \mathbb{E}[q(w)]\right). \tag{23}$$

We now upper bound the two terms in the RHS of (23). We start by showing that $\mathbb{E}\left[q(w)\right] \leq 3\sqrt{\lambda/k}$. Let $L_q = \max_{w \in \mathbb{R}} |q'(w)|$ be the Lipschitz constant of the function $q(w)$. It is easy to show that $L_q \leq 3\sqrt{\lambda}$. Let $z \sim \mathcal{N}(0, 1)$ be independent of $w$. Using the first order Taylor expansion yields

$$\mathbb{E}_{w,z}\left[q(w) - q\left(w + \frac{1}{\sqrt{k}}z\right)\right] \leq L_q \frac{1}{\sqrt{k}} \mathbb{E}_z\left[|z|\right] = 3\sqrt{\frac{\lambda}{k}}\sqrt{\frac{2}{\pi}} \leq 3\sqrt{\lambda/k}. \qquad (24)$$

Next, note that $(w + \frac{1}{\sqrt{k}}z) \sim N(0, 1)$. Therefore

$$\mathbb{E}_{w,z}\left[q\left(w + \frac{1}{\sqrt{k}}z\right)\right] = \int_{-\infty}^{\infty} q(v)\frac{e^{-v^2/2}}{\sqrt{2\pi}}dv$$

$$= \int_{-\infty}^{\infty} \left(\tanh(\lambda + \sqrt{\lambda}v) - \tanh^2(\lambda + \sqrt{\lambda}v)\right)\frac{e^{-v^2/2}}{\sqrt{2\pi}}dv.$$

Making a change of variables $x = \lambda + \sqrt{\lambda}v$, gives

$$\mathbb{E}_{w,z}\left[q\left(w + \frac{1}{\sqrt{k}}z\right)\right] = \int_{-\infty}^{\infty} \left(\tanh(x) - \tanh^2(x)\right)\frac{e^{-(x-\lambda)^2/2\lambda}}{\sqrt{2\pi\lambda}}dx.$$

Define $t_\lambda(x) = \left(\tanh(x) - \tanh^2(x)\right)\frac{e^{-(x-\lambda)^2/2\lambda}}{\sqrt{2\pi\lambda}}$. Note that $t_\lambda(x)$ is an absolutely integrable function which satisfies $t_\lambda(x) = -t_\lambda(-x)$ for any $\lambda$. Therefore, the above integral is equal to zero, namely $\int_{-\infty}^{\infty} t_\lambda(x)dx = 0$. Inserting this into (24) gives

$$\mathbb{E}_w\left[q(w)\right] \leq 3\sqrt{\lambda/k}. \qquad (25)$$

Next, we upper bound the first term on the RHS of Eq. (23). Denote by $f_W(w)$ the probability density function of $w \sim \mathcal{N}(0, \frac{k-1}{k})$. Then, writing the expectation explicitly gives

$$\mathbb{E}\left[\tanh(\lambda + \sqrt{\lambda}w)\right] = \int_{-\infty}^{\infty} \tanh(\lambda + \sqrt{\lambda}w)f_W(w)dw.$$

Making the change of variables $x = \lambda + \sqrt{\lambda}w$ yields

$$\mathbb{E}\left[\tanh(\lambda + \sqrt{\lambda}w)\right] = \frac{1}{\sqrt{\lambda}}\int_{-\infty}^{\infty} \tanh(x)f_W\left(\frac{x-\lambda}{\sqrt{\lambda}}\right)dx$$

$$= \frac{1}{\sqrt{\lambda}}\int_0^{\infty} \tanh(x)\left(f_W\left(\frac{x-\lambda}{\sqrt{\lambda}}\right) - f_W\left(\frac{x+\lambda}{\sqrt{\lambda}}\right)\right)dx.$$

Since $\lambda > 0$, it follows that $f_W\left(\frac{x-\lambda}{\sqrt{\lambda}}\right) - f_W\left(\frac{x+\lambda}{\sqrt{\lambda}}\right) \geq 0$, for all $x \geq 0$. Therefore, to bound this expectation it suffices to construct a function $g(x)$ such that $g(x) \geq \tanh(x)$ for any $x \geq 0$ and for which we can compute explicitly the corresponding integral. Consider the function $g(x) = x$. It is well-known that $x \geq \tanh(x)$, for all $x \geq 0$. Thus,

$$\mathbb{E}\left[\tanh(\lambda + \sqrt{\lambda}w)\right] \leq \frac{1}{\sqrt{\lambda}}\int_0^{\infty} x\left(f_W\left(\frac{x-\lambda}{\sqrt{\lambda}}\right) - f_W\left(\frac{x+\lambda}{\sqrt{\lambda}}\right)\right)dx = \frac{1}{\sqrt{\lambda}}\int_{-\infty}^{\infty} xf_W\left(\frac{x-\lambda}{\sqrt{\lambda}}\right)dx$$

Substituting $w = (x - \lambda)/\sqrt{\lambda}$, yields

$$\mathbb{E}\left[\tanh(\lambda + \sqrt{\lambda}w)\right] \leq \int_{-\infty}^{\infty} (\lambda + \sqrt{\lambda}w)f_W(w)dw = \mathbb{E}\left[\lambda + \sqrt{\lambda}w\right] = \lambda. \qquad (26)$$

Combining (23),(25) and (26) gives

$$\mathbb{E}\left[\tanh(\lambda + \sqrt{\lambda}w) - \frac{1}{2}\tanh^2\left(\lambda + \sqrt{\lambda}w\right)\right] \leq \frac{1}{2}\left(\lambda + 3\sqrt{\lambda/k}\right).$$

$\square$

**Lemma A.8.** *Consider a sequence of classification problem of the form,*

$$y \sim Unif\{\pm 1\}, \quad \boldsymbol{x}|y \sim N(y\boldsymbol{\mu}, \mathbf{I}_p),$$

*where the dimension $p \to \infty$ but $\|\boldsymbol{\mu}\| = \sqrt{\lambda}$ is fixed. Let $\hat{\boldsymbol{\mu}}_p$ be a sequence of unit norm estimators and $T_p(\boldsymbol{x}) = \text{sign}\langle\hat{\boldsymbol{\mu}}_p, \boldsymbol{x}\rangle$ the corresponding classifier. Assume that every $\epsilon > 0$*

$$\lim_{p \to \infty} \mathbb{P}(\langle\boldsymbol{\mu}/\sqrt{\lambda}, \hat{\boldsymbol{\mu}}_p\rangle > 1 - \epsilon) = 1. \tag{27}$$

*Then, the excess risk of the classifier $T_p(\boldsymbol{x}) = \text{sign}\langle\hat{\boldsymbol{\mu}}_p, \boldsymbol{x}\rangle$ tends to zero as $p$ tends to infinity.*

*Proof.* By definition, the excess risk of the classifier $T_p$ that corresponds to $\hat{\boldsymbol{\mu}}_p$ can be written as

$$\mathcal{E}(T_p) = \mathbb{P}_{\boldsymbol{\xi} \sim N(0, \mathbf{I}_p)}(\langle\hat{\boldsymbol{\mu}}_p, \boldsymbol{\mu} + \boldsymbol{\xi}\rangle < 0) - \Phi^c(\sqrt{\lambda}).$$

Since $\boldsymbol{\xi}$ is independent of $\hat{\boldsymbol{\mu}}_p$ and $\hat{\boldsymbol{\mu}}_p$ has unit norm, then $z = \langle\hat{\boldsymbol{\mu}}_p, \boldsymbol{\xi}\rangle \sim \mathcal{N}(0, 1)$, and the excess risk may be written as

$$\mathcal{E}(T_p) = \mathcal{P}(z > \langle\hat{\boldsymbol{\mu}}_p, \boldsymbol{\mu}\rangle) - \Phi^c(\sqrt{\lambda})$$

Let $\epsilon > 0$, and consider the event $\mathcal{A}_\epsilon = \{\langle\boldsymbol{\mu}/\sqrt{\lambda}, \hat{\boldsymbol{\mu}}_p\rangle > 1 - \epsilon\}$. Then,

$$\mathcal{E}(T_p) \leq \begin{cases} \Phi^c(\sqrt{\lambda}(1 - \epsilon)) - \Phi^c(\sqrt{\lambda}) & \text{with probability } \mathcal{P}(\mathcal{A}_\epsilon) \\ 1 & \text{with probability } 1 - P(\mathcal{A}_\epsilon) \end{cases}$$

Since by assumption $\mathcal{P}(\mathcal{A}_\epsilon) \to 1$ for any $\epsilon > 0$, then the excess risk tends to zero as $p \to \infty$. $\quad\square$

# B  Lower Bounds - Proofs of Results in Section 2

## B.1  Proof of Theorem 2.1

Our proof relies on Fano's inequality, and is conceptually similar to the proof of Theorem 3 in Amini & Wainwright (2009). First, note that for any sub-collection $\tilde{\mathbb{S}} \subset \mathbb{S}$, we have the following

$$\max_{S \in \mathbb{S}} \mathbb{P}\left(\hat{S} \neq S\right) \geq \frac{1}{|\tilde{\mathbb{S}}|} \sum_{S \in \tilde{\mathbb{S}}} \mathbb{P}\left(\hat{S} \neq S\right).$$

The right hand side in the above display is the error probability of an estimator $\hat{S}$, where $S$ is considered as a random variable uniformly distributed over the set $\tilde{\mathbb{S}}$. In other words, the right hand side may be written as follows,

$$\mathbb{P}(\text{error}) = \sum_{s \in \tilde{\mathbb{S}}} \mathbb{P}\left(\hat{S} \neq s \,|\, S = s\right) \cdot \mathbb{P}(S = s).$$

In our proof, we consider the following sub-collection

$$\tilde{\mathbb{S}} := \{T \in \mathbb{S} : 1, \ldots, k - 1 \in T\}, \tag{28}$$

which consists of all $k$-element subsets that contain the first $k - 1$ support indices $\{1, \ldots, k - 1\}$ and one from $\{k, \ldots, p\}$. To lower bound the probability of error, we focus on a specific class of means: given the support $S$, the mean entries have the form $\mu_j = \sqrt{\frac{\lambda}{k}}\mathbb{1}\{j \in S\}$. So, with $S$ known, $\boldsymbol{\mu}$ is deterministic and we write it as $\boldsymbol{\mu}^S$.

In the proof we consider an equivalent model of (3), whose observations are divided by $\sqrt{\lambda}$,

$$\boldsymbol{x}_i = y_i\boldsymbol{\theta}^S + \sigma\boldsymbol{\xi}_i, \quad i = 1, \ldots, L,$$

where $\boldsymbol{\theta}^S(j) = \frac{1}{\sqrt{k}}\mathbb{1}\{j \in S\}, \sigma = \frac{1}{\sqrt{\lambda}}$ and $\boldsymbol{\xi}_i \sim \mathcal{N}(0, \mathbf{I}_p)$. Since for each observation $\boldsymbol{x}_i$ we also know its corresponding label $y_i \in \{-1, 1\}$, we may consider the following transformed observations

$$\tilde{\boldsymbol{x}}_i = y_i\boldsymbol{x}_i = \boldsymbol{\theta}^S + \sigma\tilde{\boldsymbol{\xi}}_i, \quad i = 1, \ldots, L. \tag{29}$$

where $\tilde{\boldsymbol{\xi}}_i = y_i \boldsymbol{\xi}_i$ has the same distribution as $\boldsymbol{\xi}_i$. Denote by $X^L, \tilde{X}^L, Y^L$ the sets of $L$ i.i.d. samples $\{\boldsymbol{x}_i\}_{i=1}^{L}, \{\tilde{\boldsymbol{x}}_i\}_{i=1}^{L}$ and $\{y_i\}_{i=1}^{L}$, respectively. To apply Fano's lemma, we consider the joint mutual information $I\left((X^L, Y^L); S\right)$. Note that,

$$I\left(\left(X^L, Y^L\right); S\right) = I\left(\left(\tilde{X}^L, Y^L\right); S\right) = I\left(\tilde{X}^L; S\right),$$

where the last equality follows from the fact that the labels $Y^L$ are independent of the support $S$ and the observations $\tilde{X}^L$. Hence, it is enough to consider the samples $\tilde{X}^L$ from the model in (29).

Let $S$ be a subset chosen uniformly at random from $\tilde{\mathbb{S}}$. Then, from Lemma A.3, it follows that

$$\mathbb{P}\left(\text{error}\right) \geq 1 - \frac{I(\tilde{X}^L; S) + \log 2}{\log |\tilde{\mathbb{S}}|} = 1 - \frac{I(\tilde{X}^L; S) + \log 2}{\log(p - k + 1)}. \tag{30}$$

We now derive an upper bound on $I(\tilde{X}^L; S)$. First, from the relation between mutual information and conditional entropy, $I(\tilde{X}^L; S) = H(\tilde{X}^L) - H(\tilde{X}^L|S)$. By the sub-additivity of the entropy function $H(\tilde{X}^L) \leq LH(\tilde{\boldsymbol{x}})$. Also, since the samples $\tilde{\boldsymbol{x}}_i$ are conditionally independent given $S$, the joint entropy $H(\tilde{X}^L|S)$ can be expressed as

$$H(\tilde{X}^L|S) = \sum_{i \in [L]} H(\tilde{\boldsymbol{x}}_i|S) = LH(\tilde{\boldsymbol{x}}|S).$$

where $\tilde{\boldsymbol{x}}$ is a single observation from the model (29). Therefore,

$$I(\tilde{X}^L; S) \leq L\left(H(\tilde{\boldsymbol{x}}) - H(\tilde{\boldsymbol{x}}|S)\right). \tag{31}$$

By the definition of conditional entropy,

$$H(\tilde{\boldsymbol{x}}|S) = -\sum_{s \in \tilde{\mathbb{S}}} P(S = s) \int f\left(\tilde{\boldsymbol{x}}|S\right) \log f(\tilde{\boldsymbol{x}}|S) d\tilde{\boldsymbol{x}},$$

where $f(\tilde{\boldsymbol{x}}|S)$ the probability density function of a single random sample $\tilde{\boldsymbol{x}}$ given $S$. For any $S \in \tilde{\mathbb{S}}$, the vector $(\tilde{\boldsymbol{x}} \mid S)$ is a $p$-dimensional Gaussian with mean $\boldsymbol{\theta}^S$ and covariance matrix $\sigma^2 \mathbf{I}_p$. Its entropy is independent of its mean, and is given by $\frac{p}{2}\left(1 + \log(2\pi\sigma^2)\right)$. Hence,

$$H(\tilde{\boldsymbol{x}}|S) = \frac{p}{2}\left(1 + \log(2\pi) + \log(\sigma^2)\right). \tag{32}$$

The final step is to upper bound $H(\tilde{\boldsymbol{x}})$. To this end, note that $\tilde{\boldsymbol{x}}$ is distributed as a mixture of $(p-k+1)$ Gaussians, each centered at $\boldsymbol{\theta}^S$ for $S \in \tilde{\mathbb{S}}$. Let us denote its mean and covariance by $\boldsymbol{\nu}_x = \mathbb{E}\left[\tilde{\boldsymbol{x}}\right]$ and $\Sigma = \mathbb{E}\left[(\tilde{\boldsymbol{x}} - \boldsymbol{\nu}_x)(\tilde{\boldsymbol{x}} - \boldsymbol{\nu}_x)^T\right]$, respectively. By the maximum entropy property of the Gaussian distribution (Lemma A.5), and Eq. (19) for the entropy of a multivariate Gaussian, we have

$$H(\tilde{\boldsymbol{x}}) \leq H(\mathcal{N}(\boldsymbol{\nu}_x, \Sigma)) = \frac{p}{2}\left(1 + \log(2\pi)\right) + \frac{1}{2}\log\det(\Sigma). \tag{33}$$

Combining (31), (32) and (33) gives

$$I(\tilde{X}^L; S) \leq \frac{L}{2}\left(\log\det(\Sigma) - p\log\sigma^2\right). \tag{34}$$

The following lemma, proved in Appendix D, provides an upper bound for $\log\det(\Sigma)$.

**Lemma B.1.** *Let $\Sigma$ be the covariance matrix of the random vector $\tilde{\boldsymbol{x}}$ of Eq. (29), with the set $S$ uniformly distributed on $\tilde{\mathbb{S}}$ of Eq. (28). Then,*

$$\log\det(\Sigma) \leq p\log(\sigma^2) + (p - k + 1)\log\left(1 + \frac{1}{k(p - k + 1)\sigma^2}\right). \tag{35}$$

Substituting this upper bound into (34) leads to

$$I(\tilde{X}^L; S) = \frac{L}{2}(p - k + 1)\log\left(1 + \frac{1}{k(p - k + 1)\sigma^2}\right) \leq \frac{L}{2k\sigma^2} = \frac{L\lambda}{2k} \tag{36}$$

where the last inequality follows from $\log(1 + x) \leq x$, for all $x > 0$. Inserting (36) into (30), implies that a sufficient condition for the error probability to be greater than $\delta - \frac{\log 2}{\log(p-k+1)}$ is $\frac{L\lambda}{k} < 2(1 - \delta)\log(p - k + 1)$, which completes the proof.

## B.2 Proof of Theorem 2.2

To prove Theorem 2.2 we use the following lemma, proven in Appendix D.

**Lemma B.2.** *Let $\boldsymbol{x}$ be a random vector from the model (3), with a vector $\boldsymbol{\mu}^S$, where the random variable $S$ is uniformly distributed over the set $\tilde{\mathbb{S}}$ of Eq. (28), and let $I(\boldsymbol{x}; S)$ be their mutual information. Consider an asymptotic setting where $p \to \infty$ and $k/p \to \infty$. Then, for $\lambda < 1$, and for $p$ and $k$ sufficiently large with $k/p$ sufficiently small*

$$I(\boldsymbol{x}; S) \leq \frac{1}{2} \frac{\lambda^2}{k}(1 + o(1)). \tag{37}$$

First, note that for $\lambda \geq 1$, the information lower bounds proven in Theorem 2.1 and those we aim to prove in Theorem 2.2 coincide. Clearly, Theorem 2.1 which considers all possible estimators based on $\{(\boldsymbol{x}_i, y_i)\}_{i=1}^L$, includes in particular all unsupervised estimators that ignore the labels. Hence, for $\lambda \geq 1$, Theorem 2.2 follows from Theorem 2.1. Therefore, we consider the case $\lambda < 1$.

The proof, similar to that of Theorem 1, is also based on Fano's inequality. To lower bound the probability of error, we view $S$ as a subset uniformly distributed over $\tilde{\mathbb{S}}$, where $\tilde{\mathbb{S}}$ is the sub-collection of support sets defined in (28). Then, using the same arguments as in the proof of Theorem 1,

$$\max_{S \in \mathbb{S}} \mathbb{P}\left(\hat{S} \neq S\right) \geq \frac{1}{|\tilde{\mathbb{S}}|} \sum_{S \in \tilde{\mathbb{S}}} \mathbb{P}\left(\hat{S} \neq S\right) \geq 1 - \frac{I(X^n; S) + \log 2}{\log |\tilde{\mathbb{S}}|}, \tag{38}$$

where $X^n = (\boldsymbol{x}_1, ..., \boldsymbol{x}_n)$ and $I(X^n; S)$ is the mutual information between the $n$ unlabeled samples and $S$.

We now derive an upper bound for $I(X^n; S)$. The sub-additivity of the entropy function, and the fact that $\boldsymbol{x}_i$ are conditionally independent given $S = s$, imply

$$I(X^n; S) \leq n\left(H(\boldsymbol{x}) - H(\boldsymbol{x}|S)\right) = nI(\boldsymbol{x}; S), \tag{39}$$

where $\boldsymbol{x}$ is a single sample from the model (3). Hence, for $p$ and $k$ sufficiently large, with $k/p$ sufficiently small, combining (37) and (39) gives $I(X^n; S) \leq \frac{n\lambda^2}{2k}(1 + o(1))$. By Fano's bound in (38), the error probability is at least $\delta$ if $n < \frac{2(1-\delta)k}{\lambda^2} \log(p - k + 1)$.

## B.3 Proof of Theorem 2.3

Recall that the random variable $S$ is uniformly distributed over the set $\mathbb{S}$. For $X^{N_\alpha} = \{\boldsymbol{x}_i\}_{i=1}^{N_\alpha}$ and $Z^{M_\alpha} = \{\boldsymbol{z}_i\}_{i=1}^{M_\alpha}$, their combined mutual information with the random variable $S$ is

$$I(X^{N_\alpha}, Z^{M_\alpha}; S) = H(X^{N_\alpha}, Z^{M_\alpha}) - H(X^{N_\alpha}, Z^{M_\alpha}|S).$$

Since the two sets of samples $\{\boldsymbol{x}_i\}_{i=1}^{N_\alpha}$, $\{\boldsymbol{z}_j\}_{j=1}^{M_\alpha}$ are conditionally independent given $S = s$, then

$$I(X^{N_\alpha}, Z^{M_\alpha}; S) = H(X^{N_\alpha}, Z^{M_\alpha}) - H(X^{N_\alpha}|S) - H(Z^{M_\alpha}|S)$$
$$= H(X^{N_\alpha}, Z^{M_\alpha}) - N_\alpha H(x|S) - M_\alpha H(z|S). \tag{40}$$

By the sub-additivity property of the entropy function,

$$H(X^{N_\alpha}, Z^{M_\alpha}) \leq H(X^{N_\alpha}) + H(Z^{M_\alpha}) \leq N_\alpha H(x) + M_\alpha H(z). \tag{41}$$

Hence, combining (40) and (41) yields

$$I(X^{N_\alpha}, Z^{M_\alpha}; S) \leq N_\alpha \cdot I_x + M_\alpha \cdot I_z. \tag{42}$$

Combining Fano's inequality with the upper bound in (42) gives

$$\mathbb{P}\left(\hat{S} \neq S\right) \geq 1 - \frac{N_\alpha \cdot I_x + M_\alpha \cdot I_z + \log 2}{\log |\mathbb{S}|}.$$

Finally, the assumption $\max\{N \cdot I_{\boldsymbol{x}}, M \cdot I_{\boldsymbol{z}}\} < (1 - \delta) \log |\mathbb{S}|$ yields that $\mathbb{P}\left(\hat{S} \neq S\right) > \delta - \frac{\log 2}{\log |\mathbb{S}|}$

$\square$

## B.4 Proof of Corollary 2.4

To lower bound the error probability, we view $S$ as a random variable uniformly distributed over the discrete set $\tilde{\mathbb{S}}$ defined in (28). By the same arguments as in the proofs of Theorems 1 and 2,

$$\max_{S \in \mathbb{S}} \mathbb{P}\left(\hat{S} \neq S\right) \geq \mathbb{P}_{S \sim U(\tilde{\mathbb{S}})}\left(\hat{S} \neq S\right).$$

We apply Theorem 2.3, with the set $\boldsymbol{x}_i$ of i.i.d. unlabeled samples from (3), and the second set $z_i = (\boldsymbol{x}_i, y_i)$ of i.i.d. labeled samples from model (3).

Next, in the proofs of Theorems 2.1 and 2.2 the following upper bounds for $I_{\boldsymbol{z}}$ and $I_{\boldsymbol{x}}$ were derived,

$$I_{\boldsymbol{z}} \leq \frac{\lambda}{2k}, \quad I_{\boldsymbol{x}} \leq \frac{\lambda}{2k} \min\{1, \lambda\}(1 + o(1)). \tag{43}$$

Finally, by the conditions of the Corollary, the number of labeled and unlabeled samples satisfy $L = \lfloor qL_0 \rfloor$ and $n = \lfloor (1 - q)n_0 \rfloor$, with $L_0 < \frac{2(1-\delta)k}{\lambda} \log(p - k + 1)$ and $n_0 < \frac{2(1-\delta)k}{\lambda^2} \log(p - k + 1) \max\{1, \lambda\}$. Hence, for sufficient large $p$, combining these conditions with (43) gives

$$L_0 \cdot I_{\boldsymbol{z}}, \; n_0 \cdot I_{\boldsymbol{x}} \leq (1 - \delta) \log(p - k + 1)$$

Therefore, by Theorem 2.3 the error probability is at least $\delta$. $\qquad\square$

## B.5 Proof of Theorem 2.6

Even though we analyze a SSL setting, the observed data still belongs to the additive Gaussian noise model (see Section 2 in Kunisky et al. (2022)). This key point allows to simplify the low-degree norm in our setting. Specifically, let $Z = (\boldsymbol{z}_1, \ldots, \boldsymbol{z}_{L+n})$ be the following set of random vectors, which are the noise-free underlying signals from the alternative $\mathbb{P}_{L+n}$ of Eq. (9) in the main text,

$$\boldsymbol{z}_i = \begin{cases} \boldsymbol{\mu}^S, & i \in [L], \\ y_i \boldsymbol{\mu}^S, & L < i \leq L + n. \end{cases}$$

In the equation above, $S$ is uniformly distributed on $\mathbb{S}$ (the set of all size-$k$ subsets over $p$ variables), $\boldsymbol{\mu}^S$ is a $k$-sparse vector with support $S$ and non-zero entries $\sqrt{\lambda/k}$ and $y_i$ are Rademacher random variables. Similarly, let $\tilde{Z} = (\tilde{\boldsymbol{z}}_1, \ldots, \tilde{\boldsymbol{z}}_{L+n})$ be an independent set of the underlying noise-free signals, with a possibly different support $\tilde{S}$, and independent labels $\tilde{y}_i$. Then, by Theorem 1 in Kunisky et al. (2022) the low degree norm $\|\mathcal{L}_{L+n}^D\|^2$ can be expressed as

$$\|\mathcal{L}_{L+n}^D\|^2 = \mathbb{E}_{Z, \tilde{Z}}\left[\sum_{d=0}^{D} \frac{1}{d!}\left(\sum_{i=1}^{L+n} \langle \boldsymbol{z}_i, \tilde{\boldsymbol{z}}_i \rangle\right)^d\right].$$

Inserting the expressions for $\boldsymbol{z}_i$ and $\tilde{\boldsymbol{z}}_i$ into the equation above, gives

$$\|\mathcal{L}_{L+n}^D\|^2 = \sum_{d=0}^{D} \frac{1}{d!} \mathbb{E}\left[\left(\sum_{i=1}^{L} \left\langle \boldsymbol{\mu}^S, \boldsymbol{\mu}^{\tilde{S}} \right\rangle + \sum_{i=L+1}^{L+n} \left\langle y_i \boldsymbol{\mu}^S, \tilde{y}_i \boldsymbol{\mu}^{\tilde{S}} \right\rangle\right)^d\right],$$

where the expectation is over the two random sets $S, \tilde{S}$ and over the random labels $y_i$ and $\tilde{y}_i$. Since all these random variables are independent, the right hand side above simplifies to

$$\sum_{d=0}^{D} \frac{1}{d!} \mathbb{E}\left[\left(L \left\langle \boldsymbol{\mu}^S, \boldsymbol{\mu}^{\tilde{S}} \right\rangle + \sum_{i=L+1}^{L+n} y_i \tilde{y}_i \left\langle \boldsymbol{\mu}^S, \boldsymbol{\mu}^{\tilde{S}} \right\rangle\right)^d\right]$$

$$= \sum_{d=0}^{D} \frac{1}{d!} \mathbb{E}\left[\left(\left\langle \boldsymbol{\mu}^S, \boldsymbol{\mu}^{\tilde{S}} \right\rangle \left(L + \sum_{i=L+1}^{L+n} y_i \tilde{y}_i\right)\right)^d\right].$$

Denote $R_i = y_i \tilde{y}_i$, and note that $R_i$ are Rademacher random variables, independent of $S$ and $\tilde{S}$. Thus, the expectation above can be factored into the product of two separate expectations,

$$\|\mathcal{L}_{L+n}^D\|^2 = \sum_{d=0}^D \frac{1}{d!} \mathbb{E}_{S,\tilde{S}} \left[ \left\langle \boldsymbol{\mu}^S, \boldsymbol{\mu}^{\tilde{S}} \right\rangle^d \right] \mathbb{E}_{\{R_j\}_j} \left[ \left( L + \sum_{i=L+1}^{L+n} R_i \right)^d \right]. \tag{44}$$

We now separately analyze each of these two expectations, starting from the second one. By the Binomial formula,

$$\mathbb{E} \left[ \left( L + \sum_{i=L+1}^{L+n} R_i \right)^d \right] = \sum_{\ell=0}^d \binom{d}{\ell} L^{d-\ell} \mathbb{E} \left[ \left( \sum_{i=L+1}^{L+n} R_i \right)^\ell \right].$$

Note that for any odd integer $\ell$, the $\ell$-th moment of the Rademacher's sum is zero. Therefore,

$$\mathbb{E} \left[ \left( L + \sum_{i=L+1}^{L+n} R_i \right)^d \right] = \sum_{\ell=0}^{\lfloor d/2 \rfloor} \binom{d}{2\ell} L^{d-2\ell} \mathbb{E} \left[ \left( \sum_{i=L+1}^{L+n} R_i \right)^{2\ell} \right].$$

As analyzed in Löffler et al. (2022, pg. 1274)

$$\mathbb{E} \left[ \left( \sum_{i=L+1}^{L+n} R_i \right)^{2\ell} \right] \le n^\ell (2\ell-1)!!,$$

where $(2\ell-1)!! = (2\ell-1)(2\ell-3) \cdots 3 \cdot 1 = \frac{(2\ell)!}{2^\ell \ell!}$. Thus,

$$\mathbb{E} \left[ \left( L + \sum_{i=L+1}^{L+n} R_i \right)^d \right] \le L^d \sum_{\ell=0}^{\lfloor d/2 \rfloor} \frac{d!}{(d-2\ell)! \, \ell!} \left( \frac{n}{2L^2} \right)^\ell = L^d \sum_{\ell=0}^{\lfloor d/2 \rfloor} \binom{d}{\ell} \left( \frac{n}{2L^2} \right)^\ell \frac{(d-\ell)!}{(d-2\ell)!}.$$

Since $\frac{(d-\ell)!}{(d-2\ell)!} = (d-\ell) \cdots (d-2\ell+1) \le d^\ell$, then

$$\mathbb{E} \left[ \left( L + \sum_{i=L+1}^{L+n} R_i \right)^d \right] \le L^d \sum_{\ell=0}^{\lfloor d/2 \rfloor} \binom{d}{\ell} \left( \frac{nd}{2L^2} \right)^\ell$$

$$\le L^d \sum_{\ell=0}^d \binom{d}{\ell} \left( \frac{nd}{2L^2} \right)^\ell = L^d \left( 1 + \frac{nd}{2L^2} \right)^d = \left( L + \frac{nd}{2L} \right)^d. \tag{45}$$

Next, we analyze the first expectation $\mathbb{E}_{S,\tilde{S}} \left[ \langle \boldsymbol{\mu}^S, \boldsymbol{\mu}^{\tilde{S}} \rangle^d \right]$ in (44). Recall that $\mu_j^S = \sqrt{\frac{\lambda}{k}} \mathbb{1}\{j \in S\}$. Hence, $\langle \boldsymbol{\mu}^S, \boldsymbol{\mu}^{\tilde{S}} \rangle = \frac{\lambda}{k} |S \cap \tilde{S}|$. Denote by $G = |S \cap \tilde{S}|$ the size of the overlap between the sets. Then $G$ is a hypergeometric random variable with the following probability distribution, for $0 \le m \le k$,

$$\mathbb{P}(G = m) = \binom{k}{m} \binom{p-k}{k-m} \binom{p}{k}^{-1}.$$

From (Johnson et al., 2005, pg.268) this probability is upper bounded as follows

$$\mathbb{P}(G = m) \le \binom{k}{m} \left( \frac{k}{p-k} \right)^m.$$

Therefore,

$$\mathbb{E}_{S,\tilde{S}} \left[ \langle \boldsymbol{\mu}^S, \boldsymbol{\mu}^{\tilde{S}} \rangle^d \right] = \frac{\lambda^d}{k^d} \mathbb{E} \left[ |S \cap \tilde{S}|^d \right] = \frac{\lambda^d}{k^d} \sum_{m=0}^k m^d \, \mathbb{P}(G = m) \le \frac{\lambda^d}{k^d} \sum_{m=0}^k m^d \binom{k}{m} \left( \frac{k}{p-k} \right)^m. \tag{46}$$

Inserting (45) and (46) into (44) gives

$$\|\mathcal{L}_{L+n}^D\|^2 \le \sum_{d=0}^D \frac{\lambda^d}{d! k^d} \left( L + \frac{nd}{2L} \right)^d \sum_{m=0}^k m^d \binom{k}{m} \left( \frac{k}{p-k} \right)^m.$$

In the above expression, since $d \leq D$, we may upper bound $nd/2L$ by $nD/2L$. Furthermore, changing the order of summation between the two sums above, gives

$$\|\mathcal{L}_{L+n}^D\|^2 \leq \sum_{m=0}^k \binom{k}{m} \left(\frac{k}{p-k}\right)^m \sum_{d=0}^D \frac{1}{d!} \left(m\left(\frac{L\lambda}{k} + \frac{n\lambda D}{2Lk}\right)\right)^d$$

$$\leq \sum_{m=0}^k \binom{k}{m} \left(\frac{k}{p-k}\right)^m \exp\left(m\left(\frac{L\lambda}{k} + \frac{n\lambda D}{2Lk}\right)\right).$$

According to the conditions of the Theorem, $L = \lfloor \frac{2\beta k}{\lambda} \log(p-k)\rfloor$ and $n = \lfloor c_2 \frac{k^\gamma}{\lambda^2}\rfloor$ for some $c_2 > 0$ and $\gamma < 2$. Hence, for sufficiently large $p$, $L > \frac{\beta k}{\lambda} \log(p-k)$. Then, inserting these values into the above gives

$$\|\mathcal{L}_{L+n}^D\|^2 \leq \sum_{m=0}^k \binom{k}{m} \left(\frac{k}{p-k}\right)^m \exp\left(m\left(2\beta\log(p-k) + \frac{c_2 k^\gamma D}{2\beta k^2 \log(p-k)}\right)\right).$$

Setting $D = (\log(p-k))^2$, yields

$$\|\mathcal{L}_{L+n}^D\|^2 \leq \sum_{m=0}^k \binom{k}{m} \left(\frac{k}{p-k}\right)^m \exp\left(m\log(p-k)\left(2\beta + \frac{c_2}{2\beta}\frac{1}{k^{2-\gamma}}\right)\right).$$

Since $\gamma < 2$, for any fixed $\epsilon > 0$ it follows that $\frac{c_2 k^\gamma}{2\beta k^2} \leq \epsilon$ for sufficiently large $k$. Therefore

$$\exp\left(m\log(p-k)\left(2\beta + \frac{c_2}{2\beta}\frac{1}{k^{2-\gamma}}\right)\right) \leq \exp\left(m\log(p-k)(2\beta + \epsilon)\right) = (p-k)^{m(2\beta+\epsilon)}$$

Combining the above two displays gives

$$\|\mathcal{L}_{L+n}^D\|^2 \leq \sum_{m=0}^k \binom{k}{m} \left(\frac{k}{p-k}\right)^m (p-k)^{m(2\beta+\epsilon)}$$

$$= \sum_{m=0}^k \binom{k}{m} \left(\frac{k}{(p-k)^{1-2\beta-\epsilon}}\right)^m = \left(1 + \frac{k}{(p-k)^{1-2\beta-\epsilon}}\right)^k.$$

Finally, recall that by the assumptions of the theorem, $k = \lfloor c_1 p^\alpha\rfloor$ for some $\alpha \in (0, \frac{1}{2}), c_1 > 0$ and that $\beta < \frac{1}{2} - \alpha$. Choosing $\epsilon = \frac{1}{2} - \alpha - \beta > 0$, gives

$$\|\mathcal{L}_{L+n}^D\|^2 \leq \left(1 + \frac{k}{(p-k)^{1/2+\alpha-\beta}}\right)^k.$$

Since $k = \lfloor c_1 p^\alpha\rfloor$, then as $p \to \infty$, the above behaves as

$$\left(1 + \frac{c_1}{p^{1/2-\beta}(1+o(1))}\right)^{c_1 p^\alpha}$$

Therefore, for $\beta < \frac{1}{2} - \alpha$, as $p \to \infty$, $\|\mathcal{L}_{L+n}^D\|^2 \to O(1)$. $\qquad\square$

## C  SSL Algorithm

### C.1  Proof of Theorem 3.2

We prove the theorem, assuming that LSPCA is run with the correct sparsity $k$ and with a slightly smaller screening factor $\tilde{\beta} = \beta - \tilde{\epsilon}$ for a fixed (though potentially arbitrarily small) $\tilde{\epsilon} > 0$, which implies the first stage retains a bit more than $p^{1-\beta}$ of the original $p$ variables.

Our proof relies on the following two key properties: (i) The set $S_L$ of size $\tilde{p} = \lceil p^{1-\tilde{\beta}}\rceil$, which is the output of the first step of LSPCA, contains nearly all indices of the true support; (ii) since the reduced dimension $\tilde{p} \ll n$, the leading eigenvector of PCA is asymptotically consistent, thus allowing recovery nearly all support indices.

The following lemma formally states the first property. Its proof appears in Appendix D.

**Lemma C.1.** *Let $\{(\boldsymbol{x}_i, y_i)\}_{i=1}^{L}$ be $L$ i.i.d. labeled samples from the mixture model* (3) *with a vector $\boldsymbol{\mu}$ of sparsity $k = \lfloor c_1 p^\alpha \rfloor$ and nonzero entries $\pm\sqrt{\lambda/k}$. Suppose that $L = \left\lceil \frac{2\beta k \log(p-k)}{\lambda} \right\rceil$, for some $\beta \in (0, 1-\alpha)$. Let $\tilde{\beta} = \beta - \tilde{\epsilon}$ where $\tilde{\epsilon} > 0$ is sufficiently small so that $\tilde{\beta} > 0$. Let $S_L$ be the indices of the top $\tilde{p} = \lceil p^{1-\tilde{\beta}} \rceil$ entries of the vector $\boldsymbol{w}_L$ of Eq.* (10). *Then, for any $\epsilon > 0$,*

$$\lim_{p \to \infty} \mathbb{P}\left( \frac{|S \cap S_L|}{k} \geq 1 - \epsilon \right) = 1. \tag{47}$$

*Proof of Theorem 3.2.* As described above, we run LSPCA with $\tilde{\beta} = \beta - \tilde{\epsilon}$, and denote by $S_L$ the set found by the first step of the algorithm. By Lemma C.1 this set satisfies Eq. (47). Denote by $\Sigma|_{S_L}$ and $\hat{\Sigma}|_{S_L}$ the population and sample covariance matrices restricted to the set of indices $S_L$. Note that,

$$\Sigma|_{S_L} = \boldsymbol{\mu}|_{S_L} \boldsymbol{\mu}|_{S_L}^{\top} + \mathbf{I}_{\tilde{p}}.$$

Hence, up to sign, the leading eigenvector of $\Sigma|_{S_L}$ is $\frac{\boldsymbol{\mu}|_{S_L}}{\|\boldsymbol{\mu}|_{S_L}\|}$. Denote by $\hat{\mathbf{v}}_{\mathrm{PCA}}$ the unit norm leading eigenvector of the sample covariance $\hat{\Sigma}|_{S_L}$. We now show that these two eigenvectors are close to each other. Indeed, since $\tilde{\beta} > 1 - \alpha\gamma$, we have $n/\tilde{p} = p^{\alpha\gamma}/\lambda^2(p^{1-\beta^*}) \to \infty$, as $p \to \infty$. Then, combining this observation with Theorem 2.3 in (Nadler, 2008), implies that with probability tending to 1,

$$\lim_{p \to \infty} \left| \left\langle \hat{\mathbf{v}}_{\mathrm{PCA}}, \frac{\boldsymbol{\mu}|_{S_L}}{\|\boldsymbol{\mu}|_{S_L}\|} \right\rangle \right| = 1.$$

Since $\boldsymbol{\mu} \in \mathbb{R}^p$ is a $k$-sparse with non-zero entries $\pm\sqrt{\frac{\lambda}{k}}$, Eq. (47) implies that $\|\boldsymbol{\mu}|_{S_L}\| \to \sqrt{\lambda}$, as $p \to \infty$. Hence,

$$\lim_{p \to \infty} \left| \left\langle \hat{\mathbf{v}}_{\mathrm{PCA}}, \frac{\boldsymbol{\mu}|_{S_L}}{\sqrt{\lambda}} \right\rangle \right| = 1, \tag{48}$$

which implies that with the correct sign,

$$\lim_{p \to \infty} \left\| \hat{\mathbf{v}}_{\mathrm{PCA}} - \frac{\boldsymbol{\mu}|_{S_L}}{\sqrt{\lambda}} \right\| = 0.$$

From now we extend $\hat{\mathbf{v}}_{\mathrm{PCA}}$ which originally had dimension $|S_L$, to a $p$−dimensional vector with zeros in $S_L^c$. Hence, since $\|\boldsymbol{\mu}|_{S_L^c}\| \to 0$, it follows

$$\lim_{p \to \infty} \left\| \hat{\mathbf{v}}_{\mathrm{PCA}} - \frac{\boldsymbol{\mu}}{\sqrt{\lambda}} \right\| = 0. \tag{49}$$

Next, let us assume by contradiction that there exist $\epsilon_0, \delta_0 \in (0,1)$ such that for every $p \in \mathbb{N}$, with probability at least $\delta_0$

$$\frac{|S \cap \hat{S}|}{k} < 1 - \epsilon_0.$$

where $\hat{S}$ is the set of top-k coordinates of $|\hat{\mathbf{v}}_{\mathrm{PCA}}|$. Combining this assumption and Eq. (49), with probability at least $\delta_0$

$$\lim_{p \to \infty} \left\| \hat{\mathbf{v}}_{\mathrm{PCA}}|_{\hat{S}} \right\|^2 = \lim_{p \to \infty} \frac{1}{\lambda} \left\| \boldsymbol{\mu}|_{\hat{S}} \right\|^2 = \lim_{p \to \infty} \frac{1}{\lambda} \left\| \boldsymbol{\mu}|_{\hat{S} \cap S} \right\|^2 = \lim_{p \to \infty} \frac{|\hat{S} \cap S|}{k} \leq 1 - \epsilon_0, \tag{50}$$

where the last inequality follows from the above assumption and $|\mu_j| = \sqrt{\lambda/k}$, for all $j \in S$.

Next, from (47) and (49) it follows that for any subset $T$ that satisfies $S_L \cap S \subset T \subset S_L$ and $|T| = k$,

$$\lim_{p \to \infty} \left\| \hat{\mathbf{v}}_{\mathrm{PCA}}|_T \right\|^2 = \lim_{p \to \infty} \frac{1}{\lambda} \left\| \boldsymbol{\mu}|_T \right\|^2 = \lim_{p \to \infty} \frac{|S \cap S_L|}{k} = 1. \tag{51}$$

However, since $\hat{S}$ is the set of the top-k indices of $|\hat{\mathbf{v}}_{\mathrm{PCA}}|$, for any $|T| = k, T \subset S_L$

$$\left\| \hat{\mathbf{v}}^{\mathrm{PCA}}|_{\hat{S}} \right\| \geq \left\| \hat{\mathbf{v}}^{\mathrm{PCA}}|_T \right\|,$$

which is a contradiction to (50) and (51). Hence, for any $\epsilon > 0$, as $p$ tends to infinity, $\mathbb{P}\left( \frac{|S \cap \hat{S}|}{k} \geq 1 - \epsilon \right) \to 1$, which completes the first part of the proof.

The second part of the proof follows from combining (48) and Lemma A.8.

$\square$

# D  Proofs of Additional Lemmas

*Proof of Lemma B.1.* First, note that the mean $\boldsymbol{\nu}_x = \mathbb{E}\left[\tilde{\boldsymbol{x}}\right] = \mathbb{E}_S\left[\boldsymbol{\theta}^S\right]$ is given by

$$(\boldsymbol{\nu}_x)_j = \begin{cases} \frac{1}{\sqrt{k}} & 1 \le j \le k-1 \\ \frac{1}{\sqrt{k}} \frac{1}{p-k+1} & k \le j \le p. \end{cases}$$

To derive an explicit expression for the covariance matrix $\Sigma$ we use the law of total expectation

$$\Sigma = \frac{1}{p-k+1}\sum_{j=k}^{p}\mathbb{E}\left[(\tilde{\boldsymbol{x}}-\boldsymbol{\nu}_x)(\tilde{\boldsymbol{x}}-\boldsymbol{\nu}_x)^\top \mid S=s_j\right]. \tag{52}$$

where $s_j = [k-1] \cup \{j\}$ is a member of $\tilde{\mathbb{S}}$. By definition,

$$((\tilde{\boldsymbol{x}}-\boldsymbol{\nu}_x)\mid S=s_j) = \left(\boldsymbol{\theta}^{S_j}-\boldsymbol{\nu}_x\right)+\sigma\tilde{\boldsymbol{\xi}} = \frac{1}{\sqrt{k}}\boldsymbol{e}_j - \frac{1}{\sqrt{k}(p-k+1)}\boldsymbol{u}+\sigma\tilde{\boldsymbol{\xi}} \tag{53}$$

where $\boldsymbol{u} = \left[\mathbf{0}_{k-1}^\top, \mathbf{1}_{p-k+1}^\top\right]^\top$ and $\{\boldsymbol{e}_j\}_{j\in[p]}$ denote the standard basis of $\mathbb{R}^p$. Since $\tilde{\boldsymbol{\xi}}$ is independent of $S$, inserting (53) into (52) gives

$$\Sigma = \frac{1}{p-k+1}\sum_{j=k}^{p}\mathbb{E}\left[\left(\frac{1}{\sqrt{k}}\left(\boldsymbol{e}_j - \frac{1}{(p-k+1)}\boldsymbol{u}\right)+\sigma\tilde{\boldsymbol{\xi}}\right)\left(\frac{1}{\sqrt{k}}\left(\boldsymbol{e}_j - \frac{1}{(p-k+1)}\boldsymbol{u}\right)+\sigma\tilde{\boldsymbol{\xi}}\right)^\top\right]$$

$$= \frac{1}{k(p-k+1)^2}\left((p-k+1)\sum_{j=k}^{p}\boldsymbol{e}_j\boldsymbol{e}_j^T - \left(\boldsymbol{u}\sum_{j=k}^{p}\boldsymbol{e}_j^\top + \sum_{j=k}^{p}\boldsymbol{e}_j\boldsymbol{u}^\top\right)+\boldsymbol{u}\boldsymbol{u}^\top\right)+\sigma^2\mathbf{I}_p$$

Note that $\sum_{j=k}^{p}\boldsymbol{e}_j = \boldsymbol{u}$. Thus,

$$\Sigma = \frac{1}{k(p-k+1)^2}\left((p-k+1)\sum_{j=k}^{p}\boldsymbol{e}_j\boldsymbol{e}_j^T - \boldsymbol{u}\boldsymbol{u}^\top\right)+\sigma^2\mathbf{I}_p$$

$$\preceq \frac{1}{k(p-k+1)}\sum_{j=k}^{p}\boldsymbol{e}_j\boldsymbol{e}_j^T + \sigma^2\mathbf{I}_p$$

$$= \frac{1}{k(p-k+1)}\begin{bmatrix}\mathbf{0}_{(k-1)\times(k-1)} & \mathbf{0}_{(k-1)\times(p-k+1)} \\ \mathbf{0}_{(p-k+1)\times(k-1)} & \mathbf{I}_{(\mathbf{p-k+1})\times(\mathbf{p-k+1})}\end{bmatrix}+\sigma^2\mathbf{I}_p.$$

Therefore,

$$\log\det(\Sigma) \le \log\left((\sigma^2)^p\left(1+\frac{1}{k(p-k+1)\sigma^2}\right)^{p-k+1}\right)$$

$$= p\log(\sigma^2) + (p-k+1)\log\left(1+\frac{1}{k(p-k+1)\sigma^2}\right).$$

$\square$

*Proof of Lemma B.2.* By definition $I(\boldsymbol{x}; S) = H(\boldsymbol{x}) - H(\boldsymbol{x}|S)$. Hence, we first derive expressions for these two terms. Since $\boldsymbol{x}$ follows the mixture model (3), it is of the form $\boldsymbol{x} = y\boldsymbol{\mu}^S + \boldsymbol{\xi}$. Given $S=s$, $\boldsymbol{\mu}^s$ is deterministic with $\mu_j^s = \sqrt{\frac{\lambda}{k}}\mathbb{1}\{j \in s\}$. Thus, the vector $(\boldsymbol{x}|S=s)$ is distributed as a mixture of two Gaussians with centers $\pm\boldsymbol{\mu}^s$ and identity covariance matrix. Its density is

$$f(\boldsymbol{x}|S=s) = \frac{1}{(2\pi)^{p/2}}\left(\frac{e^{-\|\boldsymbol{x}-\boldsymbol{\mu}^s\|^2/2}+e^{-\|\boldsymbol{x}+\boldsymbol{\mu}^s\|^2/2}}{2}\right) = \frac{e^{-\frac{\|\boldsymbol{x}\|^2+\lambda}{2}}}{(2\pi)^{p/2}}\left(\frac{e^{-\langle\boldsymbol{x},\boldsymbol{\mu}^s\rangle}+e^{\langle\boldsymbol{x},\boldsymbol{\mu}^s\rangle}}{2}\right). \tag{54}$$

By the definition of conditional entropy,

$$H(\boldsymbol{x}|S) = -\sum_{s \in \tilde{\mathbb{S}}} P(S = s) \int f(\boldsymbol{x}|S = s) \log f(\boldsymbol{x}|S = s) d\boldsymbol{x}.$$

Given the structure of the vectors $\boldsymbol{\mu}^s$ for all $s \in \tilde{\mathbb{S}}$, all the integrals in the sum above give the same value. Therefore, it suffices to consider a single set $s_0 = \{1, \ldots, k\}$,

$$H(\boldsymbol{x}|S) = -\int f(\boldsymbol{x}|S = s_0) \log f(\boldsymbol{x}|S = s_0) d\boldsymbol{x}. \tag{55}$$

Inserting (54) into (55), gives

$$H(\boldsymbol{x}|S) = -\mathbb{E}_{\boldsymbol{x}|s_0} \left[ \log \left( \frac{e^{-\frac{\|\boldsymbol{x}\|^2 + \lambda}{2}}}{(2\pi)^{p/2}} \cosh(\langle \boldsymbol{x}, \boldsymbol{\mu}^{s_0} \rangle) \right) \right].$$

Note that for any $s \in \tilde{\mathbb{S}}$, $\mathbb{E}\left[\|\boldsymbol{x}\|^2 \mid S = s\right] = \lambda + p$. Thus,

$$H(\boldsymbol{x}|S) = C(p, \lambda) - \mathbb{E}_{\boldsymbol{x}|s_0} \left[\log \cosh(\langle \boldsymbol{x}, \boldsymbol{\mu}^{s_0} \rangle)\right]. \tag{56}$$

where $C(p, \lambda) = \lambda + \frac{p}{2}(1 + \log(2\pi))$.

Consider the following two independent random variables, $w = \frac{1}{\sqrt{k}} \sum_{j=1}^{k-1} \xi_j \sim N\left(0, \frac{k-1}{k}\right)$ and $\xi = \xi_k \sim N(0, 1)$. For $S = s_0$,

$$\langle \boldsymbol{x}, \boldsymbol{\mu}^{s_0} \rangle = \langle y\boldsymbol{\mu}^{s_0} + \boldsymbol{\xi}, \boldsymbol{\mu}^{s_0} \rangle = \lambda y + \sqrt{\lambda} w + \frac{\sqrt{\lambda}}{\sqrt{k}} \xi.$$

Inserting the above into (56) gives

$$H(\boldsymbol{x}|S) = C(p, \lambda) - \mathbb{E}\left[\log \cosh(\lambda y + \sqrt{\lambda} w + \sqrt{\lambda/k} \xi)\right],$$

where the expectation is over $w, y$ and $\xi$. Note that $w, y$ and $\xi$ are independent random variables with zero mean and symmetric distributions around zero. Further, recall that $y$ attains the values $\pm 1$ with equal probabilities. Hence, by a symmetry argument we may set $y = 1$ and take the expectation only over $w$ and $\xi$. This gives

$$H(\boldsymbol{x}|S) = \lambda + \frac{p}{2}(1 + \log 2\pi) - \mathbb{E}\left[\log \cosh\left(\lambda + \sqrt{\lambda} w + \sqrt{\lambda/k} \xi\right)\right]. \tag{57}$$

Next, we derive an expression for $H(\boldsymbol{x})$. Recall that $\boldsymbol{x}$ depends on a vector $\boldsymbol{\mu}^S$ with $S$ distributed uniformly at random from $\tilde{\mathbb{S}}$ of size $p - k + 1$. Note that $\tilde{\mathbb{S}} = \bigcup_{\ell=k}^{p} s_\ell$ where $s_\ell = [k-1] \cup \{\ell\}$. By the law of total probability

$$f(\boldsymbol{x}) = \frac{1}{|\tilde{\mathbb{S}}|} \sum_{s \in \tilde{\mathbb{S}}} f(\boldsymbol{x}|S = s) = \frac{1}{p - k + 1} \sum_{\ell=k}^{p} f(\boldsymbol{x}|S = s^\ell).$$

Using the same analysis as before, it follows that

$$f(\boldsymbol{x}) = \frac{e^{-\frac{\|\boldsymbol{x}\|^2 + \lambda}{2}}}{(2\pi)^{p/2}} \cdot \frac{1}{p - k + 1} \sum_{\ell=k}^{p} \frac{e^{-\langle \boldsymbol{x}, \boldsymbol{\mu}^{s^\ell} \rangle} + e^{\langle \boldsymbol{x}, \boldsymbol{\mu}^{s^\ell} \rangle}}{2}.$$

Hence,

$$H(\boldsymbol{x}) = -\mathbb{E}\left[\log f(\boldsymbol{x})\right] = C(p, \lambda) - \mathbb{E}\left[\log\left(\frac{1}{p - k + 1} \sum_{\ell=k}^{p} \frac{e^{-\langle \boldsymbol{x}, \boldsymbol{\mu}^{s^\ell} \rangle} + e^{\langle \boldsymbol{x}, \boldsymbol{\mu}^{s^\ell} \rangle}}{2}\right)\right] \tag{58}$$

We now simplify the expectation in (58). First, by a symmetry argument, we may assume the label that corresponds to $\boldsymbol{x}$ is simply $y = 1$. Let us simplify the inner product $\langle \boldsymbol{x}, \boldsymbol{\mu}^{s_\ell} \rangle$.

$$\langle \boldsymbol{x}, \boldsymbol{\mu}^{s_\ell} \rangle = \langle \sqrt{\lambda} \boldsymbol{\mu}^S + \boldsymbol{\xi}, \boldsymbol{\mu}^{s_\ell} \rangle = \lambda \left(\frac{k-1}{k} + \frac{\mathbb{1}\{\ell \in S\}}{k}\right) + \sqrt{\lambda} w + \frac{\sqrt{\lambda}}{\sqrt{k}} \xi_\ell.$$

Hence, the expectation above can be written as

$$\mathbb{E}\left[\log\left(\frac{1}{p-k+1}\sum_{\ell=k}^{p}\frac{e^{-\left(\lambda\frac{(k-1)}{k}+\sqrt{\lambda}w+\sqrt{\lambda/k}\xi_\ell+\frac{\lambda}{k}\mathbb{1}\{\ell\in S\}\right)}+e^{\left(\lambda\frac{k-1}{k}+\sqrt{\lambda}w+\sqrt{\lambda/k}\xi_\ell+\frac{\lambda}{k}\mathbb{1}\{\ell\in S\}\right)}}{2}\right)\right]$$

where the expectation is over $S$, $w$ and $\{\xi_\ell\}_{\ell=k}^{p}$. Since $S$ is uniformly distributed over $\tilde{\mathbb{S}}$ and for any $S = s^j$ the expectation is the same, we may thus set $S = s^k = [k]$, and take the expectation only over $w$ and $\{\xi_\ell\}_{\ell=k}^{p}$.

Next, we decompose the sum inside the logarithm as $S_1 + S_2$, where

$$S_1 = \frac{\exp\left(-\lambda\frac{k-1}{k}-\sqrt{\lambda}w\right)}{2}\cdot\frac{1}{p-k+1}\sum_{\ell=k}^{p}e^{-\sqrt{\frac{\lambda}{k}}\xi_\ell-\frac{\lambda}{k}\mathbb{1}\{\ell=k\}}$$

$$S_2 = \frac{\exp\left(\lambda\frac{k-1}{k}+\sqrt{\lambda}w\right)}{2}\cdot\frac{1}{p-k+1}\sum_{\ell=k}^{p}e^{\sqrt{\frac{\lambda}{k}}\xi_\ell+\frac{\lambda}{k}\mathbb{1}\{\ell=k\}}$$

We now analyze each of these terms in an asymptotic setting where $p, k \to \infty$ and $k = o(p)$. To this end we write the sum in $S_2$ as follows

$$\frac{1}{p-k+1}\sum_{\ell=k}^{p}e^{\sqrt{\frac{\lambda}{k}}\xi_\ell}+\frac{1}{p-k+1}e^{\sqrt{\frac{\lambda}{k}}\xi_k}\left(e^{\frac{\lambda}{k}}-1\right)$$

By the central limit theorem, asymptotically, the first sum may be written as $e^{\lambda/2k}+O_P(\frac{\sqrt{\lambda}}{\sqrt{kp}})$, which follows from the fact that $\mathbb{E}_{Z\sim\mathcal{N}(0,1)}[e^{tZ}] = e^{t^2/2}$. The second term above is $O_P(\frac{\sqrt{\lambda}}{kp})$, which is negligible w.r.t. to the previous $O_P$ term. Note that its expectation is finite and given by $\frac{e^{\lambda/2k}\left(e^{\frac{\lambda}{k}}-1\right)}{p-k+1}$. The sum $S_1$ can be analyzed similarly. In summary we obtain that

$$S_1 + S_2 = e^{\lambda/2k}\cdot\cosh\left(\lambda\frac{k-1}{k}+\sqrt{\lambda}w\right)\cdot\left(1+O_P\left(\frac{\sqrt{\lambda}}{\sqrt{kp}}\right)\right)$$

Hence, the expectation above simplifies to

$$\mathbb{E}[\log(S_1 + S_2)] = \frac{\lambda}{2k}+\mathbb{E}\left[\log\cosh\left(\lambda\frac{k-1}{k}+\sqrt{\lambda}w\right)\right]+O\left(\frac{\sqrt{\lambda}}{\sqrt{kp}}\right)$$

Inserting the above into (58) gives

$$H(\boldsymbol{x}) = C(p,\lambda)-\mathbb{E}\left[\log\cosh\left(\lambda\frac{k-1}{k}+\sqrt{\lambda}w\right)\right]+O\left(\frac{\sqrt{\lambda}}{\sqrt{kp}}\right). \qquad (59)$$

Next, we derive an upper-bound for the mutual information $I(\boldsymbol{x};S) = H(\boldsymbol{x})-H(\boldsymbol{x}|S)$. Combining (57) and (59), the constant $C(p,\lambda)$ cancels out, and we obtain

$$I(\boldsymbol{x};S) = \mathbb{E}\left[\log\cosh\left(\lambda+\sqrt{\lambda}w+\sqrt{\frac{\lambda}{k}}z\right)-\log\cosh\left(\lambda\frac{(k-1)}{k}+\sqrt{\lambda}w\right)\right]-\frac{\lambda}{2k}+O\left(\frac{\sqrt{\lambda}}{\sqrt{kp}}\right).$$

$$= \mathbb{E}\left[g_w\left(\sqrt{\frac{\lambda}{k}}z\right)-g_w\left(-\frac{\lambda}{k}\right)\right]-\frac{\lambda}{2k}+O\left(\frac{\sqrt{\lambda}}{\sqrt{kp}}\right). \qquad (60)$$

where $g_w(t) = \log\cosh\left(\lambda+\sqrt{\lambda}w+t\right)$. For future use, note that $\frac{d}{dt}g_w(t) = \tanh(\lambda+\sqrt{\lambda}w+t)$.
To upper bound $I(\boldsymbol{x};S)$ we split the expectation in (60) into two parts as follows,

$$I(\boldsymbol{x};S) = \mathbb{E}\left[g_w\left(\sqrt{\frac{\lambda}{k}}z\right)-g_w(0)\right]+\mathbb{E}\left[g_w(0)-g_w\left(-\frac{\lambda}{k}\right)\right]-\frac{\lambda}{2k}+O\left(\frac{\sqrt{\lambda}}{\sqrt{kp}}\right). \qquad (61)$$

The Taylor expansion of $g_w(t)$ is given by

$$g_w(t) = g_w(0) + t\tanh(\lambda + \sqrt{\lambda}w) + \frac{t^2}{2}(1 - \tanh^2(\lambda + \sqrt{\lambda}w)) + \frac{t^3}{3!}g_w^{(3)}(\tau_t).$$

Here $\tau_t$ is some number between $0$ and $t$, and $g_w^{(3)}(\tau_t) = -2\tanh(\lambda + \sqrt{\lambda}w + \tau_t) + 2\tanh^3(\lambda + \sqrt{\lambda}w + \tau_t)$. Note that for all $t \in \mathbb{R}$,

$$\frac{t^3}{3!}g_w^{(3)}(\tau_t) \le \frac{2|t^3|\tanh(\lambda + \sqrt{\lambda}w + \tau_t)}{3!}(1 - \tanh^2(\lambda + \sqrt{\lambda}w + \tau_t)) \le \frac{2|t^3|}{3!}.$$

Since $z, w$ are independent and $\mathbb{E}[z] = 0$, it follows that

$$\mathbb{E}\left[g_w\left(\sqrt{\frac{\lambda}{k}}z\right) - g_w(0)\right]$$

$$\le \mathbb{E}\left[\sqrt{\frac{\lambda}{k}}z\tanh\left(\lambda + \sqrt{\lambda}w\right) + \frac{\lambda}{2k}z^2\left(1 - \tanh^2\left(\lambda + \sqrt{\lambda}w\right)\right) + \frac{2\lambda^{3/2}|z^3|}{3!k^{3/2}}\right]$$

$$= \frac{\lambda}{2k}\mathbb{E}\left[1 - \tanh^2\left(\lambda + \sqrt{\lambda}w\right)\right] + O\left(\sqrt{\frac{\lambda^3}{k^3}}\right). \tag{62}$$

For the second term in (61), for any value of $w$, by the mean value theorem, it follows that

$$g_w(0) - g_w\left(-\frac{\lambda}{k}\right) = \frac{\lambda}{k}\tanh(\lambda + \sqrt{\lambda}w + \zeta_w) \le \frac{\lambda}{k}\tanh(\lambda + \sqrt{\lambda}w)$$

where $\zeta_w \in [-\lambda/k, 0]$, and the last inequality follows from the fact that $\tanh(\cdot)$ is a monotonically increasing function. Hence,

$$\mathbb{E}\left[g_w(0) - g_w\left(-\frac{\lambda}{k}\right)\right] \le \frac{\lambda}{k}\mathbb{E}\left[\tanh(\lambda + \sqrt{\lambda}w)\right]. \tag{63}$$

Hence, combining (61), (62) and (63), yields

$$I(\boldsymbol{x}; S) \le \frac{\lambda}{2k}\mathbb{E}\left[1 - \tanh^2\left(\lambda + \sqrt{\lambda}w\right)\right] + \frac{\lambda}{k}\mathbb{E}\left[\tanh(\lambda + \sqrt{\lambda}w)\right] - \frac{\lambda}{2k} + O\left(\sqrt{\frac{\lambda^3}{k^3}} + \frac{\sqrt{\lambda}}{\sqrt{kp}}\right)$$

$$= \frac{\lambda}{k}\mathbb{E}\left[\tanh(\lambda + \sqrt{\lambda}w) - \frac{1}{2}\tanh^2\left(\lambda + \sqrt{\lambda}w\right)\right] + O\left(\sqrt{\frac{\lambda^3}{k^3}} + \frac{\sqrt{\lambda}}{\sqrt{kp}}\right). \tag{64}$$

By Lemma A.7, the expectation above can be bounded as follows:

$$\mathbb{E}\left[\tanh(\lambda + \sqrt{\lambda}w) - \frac{1}{2}\tanh^2\left(\lambda + \sqrt{\lambda}w\right)\right] \le \frac{1}{2}\left(\lambda + 3\sqrt{\lambda/k}\right).$$

Hence, inserting this upper bound into (64), gives

$$I(\boldsymbol{x}; S) \le \frac{1}{2}\frac{\lambda^2}{k} + O\left(\frac{\sqrt{\lambda}}{\sqrt{kp}} + \sqrt{\lambda^3/k^3}\right) \tag{65}$$

Asymptotically, as $p, k \to \infty$ with $k/p \to 0$ then $\frac{1}{k} \gg \frac{1}{\sqrt{kp}}$. Thus, the term $\frac{1}{2}\frac{\lambda^2}{k}$ is asymptotically larger than the $O(\cdot)$ terms in the display above. Hence, the inequality (37) of the lemma follows. $\square$

*Proof of Lemma C.1.* The main idea of the proof is to show that with a suitable choice of threshold $\tau$, nearly all entries $\boldsymbol{w}_L(j)$ for $j \in S$ are above this threshold, in absolute value, whereas the number of noise magnitudes above it is smaller than $\tilde{p} - k$. Since we prove the lemma for the case of two symmetric Gaussians $\boldsymbol{\mu}_1 = -\boldsymbol{\mu}_{-1} = \boldsymbol{\mu}$, for simplicity we consider the following formula

for $\boldsymbol{w}_L = \frac{1}{L}\sum_{i=1}^L y_i \boldsymbol{x}_i$. With minor adaptations one can also prove the Lemma with the original formula (10) of $\boldsymbol{w}_L$.

Fix $\epsilon > 0$, and let $\mathcal{A}_L$ denote the event that $|S_L \cap S| \geq k(1-\epsilon)$. For any threshold $\tau$ define the following two events,

$$\tilde{\mathcal{B}}(\tau) = \left\{ \sum_{j \in S} \mathbb{1}\{|\boldsymbol{w}_L(j)| > \tau\} > k(1-\epsilon) \right\},$$

and

$$\tilde{\mathcal{C}}(\tau) = \left\{ \sum_{j \notin S} \mathbb{1}\{|\boldsymbol{w}_L(j)| > \tau\} < \tilde{p} - k \right\}.$$

By their definition, it follows that for any $\tau > 0$

$$\tilde{\mathcal{B}}(\tau) \cap \tilde{\mathcal{C}}(\tau) \subseteq \mathcal{A}_L.$$

Furthermore, note that the two events $\tilde{\mathcal{B}}(\tau)$ and $\tilde{\mathcal{C}}(\tau)$ are independent. Hence, to prove that $\mathbb{P}(\mathcal{A}_L) \to 1$, it suffices to prove that for a suitable threshold $\tau$,

$$\mathbb{P}\left(\tilde{\mathcal{B}}(\tau) \cap \tilde{\mathcal{C}}(\tau)\right) = \mathbb{P}\left(\tilde{\mathcal{B}}(\tau)\right) \cdot \mathbb{P}\left(\tilde{\mathcal{C}}(\tau)\right) \to 1 \tag{66}$$

In other words, it suffices to show that each of these events occurs with probability tending to one.

We start by showing that $\Pr[\tilde{\mathcal{B}}(\tau)] \to 1$. First, let us define an even simpler event, with the absolute value removed,

$$\mathcal{B}(\tau) = \left\{ \sum_{j \in S} \mathbb{1}\{\mathrm{sign}(\mu_j)\boldsymbol{w}_L(j) > \tau\} > k(1-\epsilon) \right\}$$

Clearly $\mathcal{B}(\tau) \subset \tilde{\mathcal{B}}(\tau)$ and thus it suffices to show that $\mathbb{P}(\mathcal{B}(\tau)) \to 1$ as $p \to \infty$.

To this end, we consider a threshold of the form $\tau = \sqrt{\frac{\lambda}{k}}T$, with the value of $T$ specified below. By the sparse mixture model (3), at support coordinates,

$$\mathrm{sign}(\mu_j)\boldsymbol{w}_L(j) = \sqrt{\frac{\lambda}{k}} + \frac{\mathrm{sign}(\mu_j)}{\sqrt{L}}\xi_j$$

where $\xi_j \sim \mathcal{N}(0,1)$. Inserting this expression with $L \geq \frac{2\beta k \log(p-k)}{\lambda}$ into the above, and suppressing the dependence on $\tau$ in $\mathcal{B}(\tau)$, gives

$$\begin{aligned} \mathbb{P}(\mathcal{B}) &\geq & \mathbb{P}\left( \sum_{j \in S} \mathbb{1}\left\{ 1 + \sqrt{\frac{1}{2\beta \log(p-k)}}\xi_j > T \right\} > k(1-\epsilon) \right) \\ &=& \mathbb{P}\left( \sum_{j \in S} \mathbb{1}\{\xi_j > -(1-T)\sqrt{2\beta \log(p-k)}\} > k(1-\epsilon) \right). \end{aligned} \tag{67}$$

Next, we choose

$$T = 1 - \frac{1}{(2\beta \log(p-k))^{1/4}}, \tag{68}$$

and define

$$q_1 = \mathbb{P}\left( N(0,1) > -(2\beta \log(p-k))^{1/4} \right).$$

Since the $\xi_j$'s are all independent and $|S| = k$, with this choice of $T$, Eq. (67) simplifies to

$$\mathbb{P}(\mathcal{B}) \geq \mathbb{P}(Bin(k, q_1) > k(1-\epsilon))$$

Note that $\lim_{p\to\infty} q_1 = 1$. Thus, for sufficiently large $p$, it holds that $q_1(1 - \epsilon/2) > 1 - \epsilon$. Hence, instead of the right hand side above we may bound $\mathbb{P}(Bin(k, q_1) > kq_1(1 - \epsilon/2))$. Indeed, by Chernoff's bound (17),

$$\mathbb{P}(\mathcal{B}) \geq 1 - e^{\frac{\epsilon^2 k q_1}{8}}$$

which tends to one as $p, k \to \infty$.

Next, we show that the second term in Eq. (66), $\mathbb{P}(\tilde{\mathcal{C}})$, also tends to one as $p \to \infty$. First of all, since $k = \lfloor c_1 p^\alpha \rfloor$ and $\alpha < 1 - \beta < 1 - \tilde{\beta}$, then $k \ll p^{1-\tilde{\beta}}$, and thus $\tilde{p} - k \ll \tilde{p}/2$. Hence, for $p$ sufficiently large, we may instead consider the following event

$$\mathcal{C}(\tau) = \left\{\sum_{j \notin S} \mathbb{1}\{|\boldsymbol{w}_L(j)| > \tau\} < \frac{\tilde{p}}{2}\right\} = \left\{Bin(p - k, q_2(\tau)) < \frac{\tilde{p}}{2}\right\}$$

where $q_2(\tau) = 2\Phi^c(\sqrt{L}\tau)$. Clearly $\mathcal{C}(\tau) \subset \tilde{\mathcal{C}}(\tau)$, and we now prove that with $\tau = \sqrt{\frac{\lambda}{k}}T$, and $T$ given in (68) $\mathbb{P}(\mathcal{C}(\tau)) \to 1$, by applying a Chernoff bound.

To this end, we write

$$\frac{\tilde{p}}{2} = q_2(p - k)(1 + \delta)$$

where $\delta = \frac{\tilde{p}}{2(p-k)}\frac{1}{q_2} - 1$.

To use Chernoff's inequality we first need to show that $\delta \geq 0$. Indeed, as $p \to \infty$, with $\tau = \sqrt{\frac{\lambda}{k}}T$, and using Lemma A.1 which bounds the tail function $\Phi^c$,

$$\begin{aligned}\lim_{p\to\infty}(\delta + 1) &= \lim_{p\to\infty}\frac{p^{1-\tilde{\beta}}}{2(p-k)}\frac{1}{2\Phi^c(T\sqrt{2\beta \log(p-k)})} \\ &\geq \frac{\sqrt{\pi\beta}}{2}\lim_{p\to\infty}\frac{T\sqrt{\log(p-k)}}{(p-k)^{\tilde{\beta}}\exp(-T^2\beta\log(p-k))} = \frac{\sqrt{\pi\beta}}{2}\lim_{p\to\infty}\frac{T\sqrt{\log(p-k)}}{(p-k)^{\tilde{\beta}-\beta T^2}} \quad (69)\end{aligned}$$

Note that for sufficiently large $p$,

$$\tilde{\beta} - \beta T^2 = \beta(1 - T^2) - \tilde{\epsilon} = \frac{\beta}{(2\beta\log(p-k))^{1/4}}\left(2 - \frac{1}{(2\beta\log(p-k))^{1/4}}\right) - \tilde{\epsilon} < \frac{2\beta}{(2\beta\log(p-k))^{1/4}} - \tilde{\epsilon} < 0.$$

Combining the above and (69) gives

$$\lim_{p\to\infty}(\delta + 1) \geq \frac{\sqrt{\pi\beta}}{2}\lim_{p\to\infty}T\sqrt{\log(p-k)} = \infty.$$

Since $\tilde{\beta} < \beta$, for sufficiently large $p$, and $T$ given in (68) it follows that $\tilde{\beta} - \beta T^2 < 0$. Therefore, for sufficiently large $p$, it follows that $\delta > 0$. By Chernoff's bound (16)

$$\mathbb{P}\left(\mathcal{C}\left(\sqrt{\frac{\lambda}{k}}T\right)\right) = \mathbb{P}(Bin(p - k, q_2) < q_2(p - k)(1 + \delta)) \geq 1 - e^{-\frac{\delta^2(p-k)q_2}{2+\delta}}. \quad (70)$$

We now prove that the term in the exponent tends to infinity. First, note that since $\delta \to \infty$ then $\delta = \frac{\tilde{p}}{2(p-k)q_2} - 1 \geq \frac{\tilde{p}}{4(p-k)q_2}$. Hence,

$$\lim_{p\to\infty}\frac{\delta^2(p-k)q_2}{2+\delta} = \lim_{p\to\infty}\delta(p-k)q_2 \geq \lim_{p\to\infty}\frac{\tilde{p}}{4(p-k)q_2}(p-k)q_2 = \infty.$$

Combining the above and (70) gives $\lim_{p\to\infty}\mathbb{P}\left(\mathcal{C}\left(\sqrt{\frac{\lambda}{k}}T\right)\right) = 1$ which completes the proof.

$\square$

# E  The MLE in the Supervised Setting

The following proposition states the form of the maximum likelihood estimator (MLE) for the support $S$ in a SL setting, where the sparsity $k$ is known, and the non-zero entries of $\boldsymbol{\mu}$ have magnitude $\pm\sqrt{\lambda/k}$.

**Proposition E.1.** *Let $\{(\boldsymbol{x}_i, y_i)\}_{i=1}^{L}$ be $L$ i.i.d. labeled samples from model (3) where $\boldsymbol{\mu}$ is $k$-sparse with non-zero entries $\pm\sqrt{\lambda/k}$, and let $\boldsymbol{w}_L = \frac{1}{L}\sum_{i\in[L]} y_i\boldsymbol{x}_i$. Assuming the sparsity $k$ is known, the MLE for $S = supp(\boldsymbol{\mu})$ is given by the indices corresponding to the top-$k$ magnitudes of $\boldsymbol{w}_L$.*

*Proof.* Under our assumptions, the set of all possible vectors $\boldsymbol{\mu}$ has a one-to-one mapping to a support set $S \in \mathbb{S}$ and a vector $D \in \{-1, 1\}^k$ containing the signs of the $k$ non-zero entries of $\boldsymbol{\mu}$. We thus denote $\theta = (S, D)$, and $\boldsymbol{\mu}^\theta$ by

$$\mu_j^\theta = \begin{cases} \frac{\sqrt{\lambda}}{\sqrt{k}} D_j, & j \in S, \\ 0, & j \notin S. \end{cases}$$

Let us denote by $p_{X,Y}(\boldsymbol{x}, y; \theta)$ the joint probability density function of a single sample $(\boldsymbol{x}, y)$ from the model (3) with parameter $\theta$. Since $y$ is a Rademacher random variable with a distribution not dependent on $\theta$, we may write

$$p_{X,Y}(\boldsymbol{x}, y; \theta) = p_{X|Y}(\boldsymbol{x}|y; \theta) \, p_Y(y) = \tfrac{1}{2} p_{X|Y}(\boldsymbol{x}|y; \theta).$$

Since $\boldsymbol{x}|y \sim \mathcal{N}(y\boldsymbol{\mu}, \mathbf{I}_p)$, the conditional density $p_{X|Y}(\boldsymbol{x}|y; \theta)$ simplifies to

$$p_{X,Y}(\boldsymbol{x}|y, \theta) = \frac{1}{(2\pi)^{p/2}} \exp\left(-\|\boldsymbol{x} - y\boldsymbol{\mu}^\theta\|^2/2\right). \tag{71}$$

By definition, the maximum-likelihood estimator of $\theta$ is given by

$$\hat{\theta}^{(\mathrm{ML})} = \arg\max_\theta \sum_{i=1}^{L} \log p_{X,Y}(\boldsymbol{x}, y; \theta),$$

Inserting (71) into the above, and using $\|\boldsymbol{\mu}^\theta\| = \sqrt{\lambda}$ (fixed for all $\theta$), gives

$$\hat{\theta}^{(\mathrm{ML})} = \arg\max_\theta \sum_{i=1}^{L}\left(-\|\boldsymbol{x}_i - y_i\boldsymbol{\mu}^\theta\|^2\right) = \arg\max_{\hat{\theta}} \sum_{i=1}^{L}\langle y_i\boldsymbol{x}_i, \boldsymbol{\mu}^\theta\rangle = \arg\max_\theta \langle \boldsymbol{w}_L, \boldsymbol{\mu}^\theta\rangle.$$

Therefore, the maximum value of $\langle \boldsymbol{w}_L, \boldsymbol{\mu}^\theta\rangle$ is obtained for $\hat{S}^{(\mathrm{ML})}$ being the set of indices corresponding to the $k$ largest magnitude entries of $\boldsymbol{w}_L$, and $\hat{D}^{(\mathrm{ML})} = \{\mathrm{sign}(\boldsymbol{w}_L)_j : j \in \hat{S}^{(\mathrm{ML})}\}$. $\square$

The next proposition shows that with sufficient number of labeled samples the MLE for $S$ has significant overlap with the true support set $S$.

**Proposition E.2.** *Let $\mathcal{D}_L = \{(\boldsymbol{x}_i, y_i)\}_{i=1}^{L}$ be a set of $L$ i.i.d. labeled samples from the model (3) with a $k$-sparse vector $\boldsymbol{\mu}$ whose non-zero entries are $\pm\sqrt{\lambda/k}$. Assume that for some $\alpha \in (0,1)$, $k = \lfloor c_1 p^\alpha \rfloor$ and that $L = \left\lceil \frac{2\beta k \log(p-k)}{\lambda} \right\rceil$, for some $\beta \in (0,1)$. Let $S_L$ be the indices of the $k$ largest magnitudes of the vector $\boldsymbol{w}_L = \frac{1}{L}\sum_{i\in[L]} y_i\boldsymbol{x}_i$. If $\beta > 1 - \alpha$, then for every $\epsilon \in (0,1)$,*

$$\lim_{p\to\infty} \mathbb{P}\left(\frac{|S \cap S_L|}{k} > 1 - \epsilon\right) = 1.$$

*Proof.* Fix $\epsilon > 0$, and let $\mathcal{A}_L$ denote the event that $|S_L \cap S| \geq k(1 - \epsilon)$. For any threshold $\tau$ define the following two events,

$$\tilde{\mathcal{B}}(\tau) = \left\{\sum_{j\in S} \mathbb{1}\{|\boldsymbol{w}_L(j)| > \tau\} > k(1 - \epsilon)\right\}, \quad \tilde{\mathcal{C}}(\tau) = \left\{\sum_{j\notin S} \mathbb{1}\{|\boldsymbol{w}_L(j)| > \tau\} < k\epsilon\right\}.$$

By their definition, it follows that for any $\tau > 0$, $\tilde{\mathcal{B}}(\tau) \cap \tilde{\mathcal{C}}(\tau) \subset \mathcal{A}_L$. Since the two events $\tilde{\mathcal{B}}(\tau)$ and $\tilde{\mathcal{C}}(\tau)$ are independent,

$$\mathbb{P}(\mathcal{A}_L) \geq \mathbb{P}\left(\tilde{\mathcal{B}}(\tau) \cap \tilde{\mathcal{C}}(\tau)\right) = \mathbb{P}\left(\tilde{\mathcal{B}}(\tau)\right) \cdot \mathbb{P}\left(\tilde{\mathcal{C}}(\tau)\right) \tag{72}$$

Hence, for $\mathbb{P}(\mathcal{A}_L) \to 1$, it suffices that for a suitable threshold $\tau$, both probabilities on the right hand side tend to one, We start with $\Pr(\tilde{\mathcal{B}}(\tau))$. To this end, we define an even simpler event, with the absolute value removed,

$$\mathcal{B}(\tau) = \left\{ \sum_{j \in S} \mathbb{1}\{\text{sign}(\mu_j)\boldsymbol{w}_L(j) > \tau\} > k(1 - \epsilon) \right\}$$

Clearly $\mathcal{B}(\tau) \subset \tilde{\mathcal{B}}(\tau)$ and thus it suffices to show that $\mathbb{P}(\tilde{\mathcal{B}}(\tau)) \to 1$ as $p \to \infty$. By the sparse mixture model (3), for $j \in S$,

$$\text{sign}(\mu_j)\boldsymbol{w}_L(j) = \sqrt{\frac{\lambda}{k}} + \frac{\text{sign}(\mu_j)}{\sqrt{L}}\xi_j,$$

where $\xi_j \sim \mathcal{N}(0, 1)$. Combining a threshold value $\tau = \sqrt{\frac{\lambda}{k} \frac{1 - \alpha + \beta}{2\beta}}$ and the assumption that $L \geq \frac{2\beta k \log(p-k)}{\lambda}$ with this expression, gives that

$$\mathbb{P}(\mathcal{B}(\tau)) \geq \mathbb{P}\left( \sum_{j \in S} \mathbb{1}\left\{ \xi_j > -\left(\sqrt{\beta} - \sqrt{\frac{1 - \alpha + \beta}{2}}\right)\sqrt{2 \log p} \right\} > k(1 - \epsilon) \right). \tag{73}$$

Let $q_1$ be the probabiity of each event in the above sum,

$$q_1 = \mathbb{P}\left( N(0, 1) > -\left(\sqrt{\beta} - \sqrt{\frac{1 - \alpha + \beta}{2}}\right)\sqrt{2 \log p} \right).$$

Since the $\xi_j$'s are all independent and $|S| = k$, with this choice of $\tau$, Eq. (73) simplifies to

$$\mathbb{P}(\mathcal{B}(\tau)) \geq \mathbb{P}(Bin(k, q_1) > k(1 - \epsilon)).$$

Note that since $\beta > 1 - \alpha$, then $\lim_{p \to \infty} q_1 = 1$. Thus, for sufficiently large $p$, it holds that $q_1(1 - \epsilon/2) > 1 - \epsilon$. Hence, the right hand side above may be bounded by $\mathbb{P}(Bin(k, q_1) > kq_1(1 - \epsilon/2))$. Indeed, by Chernoff's bound (17), $\mathbb{P}(\mathcal{B}(\tau)) \to 1$ as $p, k \to \infty$, since

$$\mathbb{P}(\mathcal{B}) \geq 1 - e^{-\frac{\epsilon^2 k q_1}{8}}.$$

Next, we show that the second term in Eq. (72), $\mathbb{P}(\tilde{\mathcal{C}})$, also tends to one as $p \to \infty$. First, note that

$$\tilde{\mathcal{C}}(\tau) = \{Bin(p - k, q_2(\tau)) < k\epsilon\}$$

where $q_2(\tau) = 2\Phi^c(\sqrt{L}\tau) = 2\Phi^c(\sqrt{(1 - \alpha + \beta) \log(p - k)})$. We now prove that $\mathbb{P}(\tilde{\mathcal{C}}(\tau)) \to 1$, by applying a Chernoff bound. To this end, we write

$$k\epsilon = q_2(p - k)(1 + \delta)$$

where $\delta = \frac{k\epsilon}{(p-k)}\frac{1}{q_2} - 1$. To use Chernoff's inequality we first need to show that $\delta \geq 0$. Indeed, as $p \to \infty$, using Lemma A.1 which bounds the tail function $\Phi^c$,

$$\lim_{p \to \infty} (\delta + 1) \geq \lim_{p \to \infty} \frac{\epsilon k}{p - k} \frac{1}{2\Phi^c(\sqrt{(1 - \alpha + \beta) \log(p - k)})}$$

$$\geq \epsilon c_1 \sqrt{\frac{\pi(1 - \alpha + \beta)}{2}} \lim_{p \to \infty} \frac{\sqrt{\log(p - k)}}{p^{1 - \alpha} \exp(-\frac{1 - \alpha + \beta}{2} \log(p - k))}$$

$$= \epsilon c_1 \sqrt{\frac{\pi(1 - \alpha + \beta)}{2}} \lim_{p \to \infty} \frac{\sqrt{\log(p - k)}}{p^{\frac{1 - \alpha - \beta}{2}}} = \infty,$$

where the last equality follows from $\beta > 1 - \alpha$. By Chernoff's bound (16)

$$\mathbb{P}\left(\tilde{\mathcal{C}}\left(\tau\right)\right) = \mathbb{P}\left(Bin(p-k, q_2) < q_2(p-k)(1+\delta)\right) \geq 1 - e^{-\frac{\delta^2(p-k)q_2}{2+\delta}}. \tag{74}$$

We now prove that the term in the exponent tends to infinity. Since $\delta \to \infty$ it follows that

$$\lim_{p\to\infty} \frac{\delta^2(p-k)q_2}{2+\delta} = \lim_{p\to\infty} \delta(p-k)q_2 \geq \lim_{p\to\infty} \frac{(1+\delta)(p-k)q_2}{2} = \lim_{p\to\infty} \frac{k\epsilon}{2} = \infty.$$

Combining the above and (74) gives that $\lim_{p\to\infty} \mathbb{P}\left(\tilde{\mathcal{C}}\left(\tau\right)\right) = 1$, which completes the proof. $\quad\square$

The above proposition implies the following result regarding accurate classification: Let $T(\boldsymbol{x}) = \text{sign}\langle\hat{\boldsymbol{\mu}}, \boldsymbol{x}\rangle$, with $\hat{\boldsymbol{\mu}} = \boldsymbol{w}_L|_{S_L}$, be a linear classifier that is constructed using only the set of labeled samples $\mathcal{D}_L$. If $\beta > 1 - \alpha$, then combining Proposition E.2 and Lemma A.8 implies that the excess risk of $T$ tends to zero as $p \to \infty$.

### E.1 Impossibility of Classification

In this section we prove a lower bound for classification in the SL setting. To do that, we consider a slightly different model, known as the *rare and weak* model. Here the sparsity of the vector $\boldsymbol{\mu}$ is not fixed at exactly $k$. Instead the vector is generated randomly with entries $\mu_j = \sqrt{\lambda/k}B_j$, where $B_j \sim Ber(\epsilon_p)$ are i.i.d. Bernoulli random variables with $\epsilon_p = \frac{k}{p} = p^{-(1-\alpha)}$. The next theorem implies that in the red region (impossible) of figure 1, indeed there exists (approximately) $k$-sparse vectors for which any classifier would asymptotically be no better than random.

**Theorem E.3.** *Let $\mathcal{D}_L = \{(\boldsymbol{x}_i, y_i)\}_{i=1}^{L}$ be set of $L$ i.i.d. labeled samples from the rare weak model $\mathcal{N}(y\boldsymbol{\mu}, I)$ where all non-zero entries of $\boldsymbol{\mu}$ are $\pm\sqrt{\lambda/k}$. Suppose $L = \lceil\frac{2\beta k \log p}{\lambda}\rceil$ and $k \propto p^\alpha$, for some $\alpha < 1$. If $\beta < 1 - \alpha$ then the classification error of any classifier based on $\mathcal{D}_L$, tends to $1/2$ as $p \to \infty$.*

*Proof.* This proof is similar to the one of Jin (2009). First, let us denote the vector of z-scores by $\boldsymbol{z} = \frac{1}{\sqrt{L}}\sum_{i\in[L]} y_i\boldsymbol{x}_i$. Since the entries of $\boldsymbol{\mu}$ are generated independently under the rare-weak model, the entries of $\boldsymbol{z}$ are also independent and all have the same density, which we denote by $f(z)$. Given $\boldsymbol{z}$, we denote the conditional probability that the $j$-th entry contains a feature by $\eta = \mathbb{P}(j \in S|\boldsymbol{z}) = \mathbb{P}(j \in S|z_j = z)$. By Bayes theorem,

$$\eta(z) = \frac{\mathbb{P}(j \in S)f(z|j \in S)}{f(z)} = \frac{\epsilon_p\phi(z - \tau_p)}{(1 - \epsilon_p)\phi(z) + \epsilon_p\phi(z - \tau_p)},$$

where $\tau_p = \sqrt{L\lambda/k} = \sqrt{2\beta \log p}$, and $\phi$ is the density of $N(0,1)$.

From Lemmas 1,2 and 4 in Jin (2009), the misclassification error of any classifier $T$ constructed using the set $\mathcal{D}_L$, can be bounded as

$$|\mathbb{P}(T(\boldsymbol{x}) \neq y) - 1/2| < C\left(1 - (\mathbb{E}_z[H(z)])^p\right)^{1/2}, \tag{75}$$

where $H(z) = \mathbb{E}_x\left[\left(1 + \eta(z)(e^{\sqrt{\lambda/k}x - \lambda/2k} - 1)\right)^{1/2}\right]$ with $x \sim N(0,1)$, and $z \sim (1 - \epsilon_p)N(0,1) + \epsilon N(\tau_p, 1)$.

Our goal is to show that $\mathbb{E}_z[H(z)] = 1 + o(1/p)$, which implies that asymptotically the accuracy of $T$ is no better than random. First, combining that $\mathbb{E}_x\left[e^{\sqrt{\lambda/k}x - \lambda/2k} - 1\right] = 0$ and the inequality $|\sqrt{1+t} - 1 - t/2| \leq Ct^2$, for any $t > -1$, gives

$$|H(z) - 1| = \left|H(z) - 1 - \tfrac{1}{2}\mathbb{E}[\eta(z)]\mathbb{E}[e^{\sqrt{\lambda/k}x - \lambda/2k} - 1]\right|$$

$$\leq C\eta^2(z)\mathbb{E}_x\left[\left(e^{\sqrt{\lambda/k}x - \lambda/2k} - 1\right)^2\right] = C\eta^2(z)(e^{\lambda/k} - 1).$$

For sufficiently large $p$, $e^{\lambda/k} - 1 \le 2\lambda/k$. Since $k \propto p^{\alpha}$, then, $|H(z) - 1| \le \tilde{C}p^{-\alpha}\eta^2(z)$. Hence, to prove that $\mathbb{E}_z[H(z)] = 1 + o(1/p)$, it suffices to show that $\mathbb{E}_z[\eta^2(z)] = o(p^{-(1-\alpha)}) = o(\epsilon_p)$. To this end, we first note

$$\mathbb{E}[\eta^2(z)] = (1 - \epsilon_p)\,\mathbb{E}[\eta^2(w)] + \epsilon_p\,\mathbb{E}[\eta^2(w + \tau_p)] \le \mathbb{E}[\eta^2(w)] + \epsilon_p\,\mathbb{E}[\eta^2(w + \tau_p)],$$

where $w \sim N(0,1)$. Write $\mathbb{E}(\eta^2(z)) = I + II + III + IV$, where we have split each of the expectations into two separate integrals, with the split at suitably chosen values $t_1$ and $t_2$,

$$I = \int_{-\infty}^{t_1} \eta^2(w)\phi(w)dw, \quad II = \int_{t_1}^{\infty} \eta^2(w)\phi(w)dw,$$

and

$$III = \epsilon_p \int_{-\infty}^{t_2} \eta^2(w + \tau_p)\phi(w)dw, \quad IV = \epsilon_p \int_{t_2}^{\infty} \eta^2(w + \tau_p)\phi(w)dw,$$

As we see below, the following values will be suitable to derive the required bounds: $t_1 = \frac{1-\alpha+\beta}{2\beta}\tau_p$ and $t_2 = \frac{1-\alpha-\beta}{2\beta}\tau_p$.

Starting with $I$, note that

$$I = \int_{-\infty}^{t_1} \left(\frac{\epsilon_p\phi(w - \tau_p)}{(1 - \epsilon_p)\phi(w) + \epsilon_p\phi(w - \tau_p)}\right)^2 \phi(w)dw \le \int_{-\infty}^{t_1} \left(\frac{\epsilon_p\phi(w - \tau_p)}{(1 - \epsilon_p)\phi(w)}\right)^2 \phi(w)dw$$

For large enough $p$, the above can be bounded via

$$I \le 2\epsilon_p^2 \int_{-\infty}^{t_1} \frac{\phi^2(w - \tau_p)}{\phi^2(w)}\phi(w)dw = C\epsilon_p^2 \int_{-\infty}^{t_1} e^{2w\tau_p - \tau_p^2 - w^2/2}dw, \tag{76}$$

where the equality follows from the definition of $\phi(w)$. Completing the square, the above can be written as

$$I \le C\epsilon_p^2 \int_{-\infty}^{t_1} e^{-(w-2\tau_p)^2/2}\, e^{\tau_p^2}dw.$$

Changing the variable $x = w - 2\tau_p$ reads

$$I \le C\epsilon_p^2 e^{\tau_p^2} \int_{-\infty}^{t_1 - 2\tau_p} e^{-x^2/2}dx = C\epsilon_p^2 e^{\tau_p^2}\Phi^c(2\tau_p - t_1) \le C\epsilon_p^2 e^{\tau_p^2}e^{-(2\tau_p - t_1)^2/2} \tag{77}$$

Finally, since $\beta < 1 - \alpha$, it follows that $C\epsilon_p^2 e^{\tau_p^2}e^{-(2\tau_p - t_1)^2/2} = o(\epsilon_p)$.

Next, since $\eta(w) < 1$ it follows that

$$II = \int_{t_1}^{\infty} \eta^2(w)\phi(w)dz \le \int_{t_1}^{\infty} \phi(w)dw = \Phi^c(t_1) \le e^{-t_1^2/2}. \tag{78}$$

Similar to the above, under the condition $\beta < 1 - \alpha$ it holds that $e^{-t_1^2/2} = o(\epsilon_p)$.

Next, note that

$$III = \epsilon_p \int_{-\infty}^{t_2} \left(\frac{\epsilon_p\phi(w)}{(1 - \epsilon_p)\phi(w + \tau_p) + \epsilon_p\phi(w)}\right)^2 \phi(w)dw$$

$$\le \epsilon_p \int_{-\infty}^{t_2} \left(\frac{\epsilon_p\phi(w)}{(1 - \epsilon_p)\phi(w + \tau_p)}\right)^2 \phi(w)dw.$$

For large enough $p$

$$III \le C\epsilon_p \int_{-\infty}^{t_2} \left(\frac{\epsilon_p\phi(w)}{\phi(w + \tau_p)}\right)^2 \phi(w)dw = C\epsilon_p^3 \int_{-\infty}^{t_2} e^{2w\tau_p + \tau_p^2 - w^2/2}dw.$$

Completing the square reads

$$III \le C\epsilon_p^3 e^{3\tau_p^2} \int_{-\infty}^{t_2} e^{-(w-2\tau_p)^2/2}dw = C\epsilon_p^3 e^{3\tau_p^2}\Phi^c(2\tau_p - t_2) \le C\epsilon_p^3 e^{3\tau_p^2}e^{-(2\tau_p - t_2)^2/2}.$$

Since $\beta \leq 1 - \alpha$ it holds that $III = o(\epsilon_p)$.

Finally, since $\eta(w) < 1$, IV can be bounded as follows

$$IV = \epsilon_p \int_{t_2}^{\infty} \phi(w)dw = \epsilon_p \Phi^c(t_2) \leq \epsilon_p e^{-t_2^2/2}.$$

Again, by $\beta < 1 - \alpha$, $IV = o(\epsilon_p)$. $\qquad\qquad\qquad\qquad\qquad\qquad\qquad\qquad\qquad$ $\square$

