# OpenReview forum: "Semi-Supervised Sparse Gaussian Classification: Provable Benefits of Unlabeled Data"
_NeurIPS.cc/2024/Conference — NeurIPS 2024 spotlight_

### Official Review · Reviewer_6RTo · 2024-06-30

**Soundness:** 3
**Presentation:** 3
**Contribution:** 3
**Rating:** 7
**Confidence:** 4

**Summary:**

In this paper, the authors studied the problem of semi-supervised learning in a 2-class classification problem with a special distribution setting. They assumed that the samples in each class came from an isotropic Gaussian distribution with unknown mean vectors $\mu_1$ and $\mu_{-1}$. They also assumed that the vector $\Delta \mu = \mu_1 - \mu_{-1}$ is sparse, with $k << p$ non-zero elements, where $p$ is the dimension of the feature space, and $\lambda = \|\Delta\mu\|_2^2/4 = \mathcal{O}(1)$.
The claims of the paper are as follows:

* Derive information-theoretic lower bounds for exact support recovery in the semi-supervised learning (SSL) setting.

* Establish computational lower bounds for classification and support recovery in SSL as $(k, L, n, p) \to \infty$, where $L$ and $n$ are the number of labeled and unlabeled samples, respectively.

* Identify a region where SSL is computationally advantageous for classification and feature selection.

The authors derive lower bounds on the number of labeled and unlabeled samples needed to efficiently recover the support set of $\Delta\mu$ and classify data. They also propose an SSL algorithm that can efficiently recover the support set of $\Delta\mu$ and classify data, outperforming any supervised or unsupervised learning algorithm that relies solely on labeled or unlabeled samples.

**Strengths:**

* The theoretical results are interesting, and the proofs are rigorous.
* The writing is very good and easy to follow.
* The messages of the paper are very clear.

**Weaknesses:**

* It seems that the algorithm named LSPCA cannot be used as a semi-supervised algorithm on real-world data and only has theoretical value in the special setting of the problem. It would be more interesting if the authors proposed an algorithm that could be applied to real-world data, yielding good results in practice while also having strong theoretical guarantees.

**Questions:**

*1*-  In line 133, it is stated that the most difficult $k$-sparse vector with a lower bound on the absolute value of the non-zero terms is a $k$-sparse vector where its non-zero elements belong to $\{\pm\sqrt{\frac{\lambda}{k}}\}$. Is there any proof for this? Is it obvious?

*2*- I think in line 823, $\Theta^S(j)$ should be $\sqrt{\frac{\lambda}{k}}1(j\in S)$. Is that correct? If it is, I think it changes the result in some of the theorems.

*3*- I believe the lower bound in Corollary 2.4 is not tight, or at least it is not proven to be tight in the paper. One reason is the assumption that given $S$, labeled and unlabeled samples are independent.

*4*- Is there any proof that the bound in Theorem 2.3 is tight?

*5*- The proof for Theorem 2.6 is indexed as the proof of Theorem 4 in the supplementary material.

*6*- In the proof of Theorem 2.6, the distributions $\mathbb{P}$ and $\mathbb{Q}$ are different from those in equations (8) and (9). Can you please explain this? Also, why should we test between the distributions in equations (8) and (9)? It seems that the distributions in the proof make more sense.

*7*- In Theorem 3.2, is it sufficient to solve the problem just for the case where non-zero elements are $\pm\sqrt{\frac{\lambda}{k}}$?

**Limitations:**

Please check the questions.

---

> ### Author Rebuttal · Authors · 2024-08-02
>
> We thank the reviewer for the constructive comments. We address the issues raised one by one.
>
> (0) We agree that our proposed LSPCA method is tailored to our specific case of a mixture of two Gaussians. However, we believe that the insights of our work, proposing a two step SSL scheme, using the labeled data for feature screening could motivate the future development of practical SSL methods for high dimensional settings.
>
> (1) We will add an explanation near L133 why vectors with non-zero entries of size $\sqrt{\lambda/k}$
> are the most difficult.
>
> (2) Our text is correct. The reason is that we work with an equivalent model where
> the noise level is $\sigma=1/\sqrt{\lambda}$ instead of $\sigma=1$. We will clarify the text.
>
> (3) The lower bound in Corollary 2.4 is indeed not necessarily tight. We will explicitly mention this in the revision.
> Please note that given $S$, by the definition of our model, the labeled and unlabeled samples
> have statistically independent Gaussian noises, so this is not an extra assumption.
>
> (4) Regarding tightness of Theorem 2.3: Please note that its proof is based on Fano's lemma. Hence,
> whenever this lemma is not tight then the theorem would be not tight as well.
>
>
> (5) Indeed the proof of Theorem 2.6 is incorrectly referenced as proof of Theorem 4. Thanks for catching this typo. which we will correct.
>
>
> (6) In the proof of Theorem 2.6, the distibutions $\mathbb{Q}$
>  and $\mathbb{P}$ are exactly those in equations (8) and (9).
>  Critically, even in the SSL setting the distribution $\mathbb{Q}_{L+n}$ consists of pure isotropic Gaussian noise vectors. We test between (8) and (9) since we follow the machinery of the low-degree likelihood ratio framework, which provides us with lower bounds.
>   We will revise the first few sentences of the proof to better explain all of these points.
>
>
> (7) Regarding theorem 3.2, please note that it is stated for $k$ sparse vectors whose non-zero entries are $\pm \sqrt{\lambda/k}$, see second line of theorem, L302.

---

> > ### Comment · Reviewer_6RTo · 2024-08-10
> >
> > I thank the authors for their answers. They addressed my questions, so I increased my score accordingly.

---

### Official Review · Reviewer_3M9B · 2024-07-05

**Soundness:** 4
**Presentation:** 4
**Contribution:** 4
**Rating:** 9
**Confidence:** 5

**Summary:**

The authors identify a regime where semi-supervised learning has computational advantage over (purely) supervised or unsupervised learning. They propose an algorithm that achieves it and demonstrates the efficacy of the algorithms in simulations.

**Strengths:**

- Extremely well-written paper
- The theoretical results I believe will become landmark results in the area of semi-supervised learning
- The literature review is thorough

**Weaknesses:**

- What is the answer to the question on L59, namely "On the computational side, is there a computational-statistical gap in SSL?" For the white "hole" in Figure 1, the computational-statistical gap remains, is this correct? It would be good to state this explicitly.
- Also, it would be nice to add references to the results in Section 2 to the paragraph starting on L86.
- It would be nice to rigorously explain the reduction on Eqn (3), even if in the appendix.

**Questions:**

- Do you have a conjecture for the information theory lower bound in the white "hole" region?
- Do you have any insights as to how one might analyze LS^2PCA?

**Limitations:**

Yes

---

> ### Author Rebuttal · Authors · 2024-08-02
>
> We thank the reviewer for the positive comments and encouraging feedback.
>
> Regarding the question in L59
> "On the computational side, is there a computational-statistical gap in SSL?"
> The answer is yes. We will modify the text in lines L70-76 to explicitly mention that by adding too few labeled samples, there is still a computational statistical gap (the orange region in Figure 1).
>
> Regarding the white "hole" in Figure 1, we do not know if the computational-statistical gap remains.
> We conjectured that the gap still remains in this region. See the text L107-8.
>
> We will revise the paragraph starting on L86 to clarify what are novel results and include references to the relevant theorems that appear later on in the manuscript.
>
> Regarding the reduction in Eq. (3), we will clarify this reduction in the revised version.
>
> Regarding an information theory lower bound in the white "hole" region:
> From an information-theoretic perspective, for $\gamma>1$, there exists a non-polynomial-time algorithm that succeeds in recovering the support, that uses only the unlabeled samples.
> Hence, since the white "hole" region is a sub-region of $\gamma>1$,  the same holds for this region as well.
>
> Regarding an analysis of \texttt{LS$^2$PCA}, please note that this would depend on the specific
> sparse-PCA method employed. In our manuscript we employed the IPU method of Tian el al. We currently do not have insight how to analyze it in our context. This is an intersting question for future work.

---

> > ### Comment · Reviewer_3M9B · 2024-08-12
> >
> > Thank you for your detailed responses. I will keep my score.

---

### Official Review · Reviewer_uoYb · 2024-07-12

**Soundness:** 3
**Presentation:** 3
**Contribution:** 3
**Rating:** 7
**Confidence:** 4

**Summary:**

This paper studies the classical problem of clustering Gaussians with sparse means. While this problem was previously considered under the unsupervised (Sparse PCA) and supervised settings, the main innovation of the authors is to identify a phase where labeled and unlabeled data can be used together to estimate the cluster means, even when this task would be impossible using only either of them.

This semi-supervised algorithm is complemented by various informational and computational lower bounds; in particular, the authors establish the existence of an information-computation gap for the detection task in the semi-supervised setting.They also provide numerical simulations for their algorithm, showing how to improve it by using out-of-the-box sparse-PCA algorithms in its second step.

**Strengths:**

I found this paper very interesting to read; it considers a very interesting and natural question, and fills most of the blank parts of the phase diagram with novel results. The main achievement of the paper, the SSL algorithm, is both completely new and very easy to understand.

**Weaknesses:**

I think the exposition of the paper could be slightly improved. In particular:
- it would be nice to mention which part of Figure 1 comes from which theorem, and which are novel to the paper;
- Theorem 2.3 and Corollary 2.4 are presented very dryly, and it is hard to understand exactly what they aim to prove (in particular because I'm not fully sure when $\alpha$ refers to an arbitrary constant or the sparsity level $\log_p(k)$).
- I didn't exactly get the point of Theorem 2.1: it seems to give a lower bound at $\beta = 1/2$ instead of $1 - \alpha$, which is weaker in the regime $\alpha \in (0, 1/2)$. What does it bring compared to the results of Donoho and Jin ?

**Questions:**

- You conjecture in the paper that the white region is actually computationally hard; do you have any (informal) arguments to support this conjecture ?

---

> ### Author Rebuttal · Authors · 2024-08-02
>
> We thank the reviewer for the constructive comments.
>
> We agree with the referee regarding context of figure 1. We will add that the red and green regions follow from previous works, whereas the orange and blue are our novel contributions. We also reference the relevant theorems for these.
>
> Regarding Theorem 2.3 and Corollary 2.4, we will rewrite it with $\alpha$ replaced by some other symbol, say $q\in(0,1)$ to avoid possible confusion with the sparsity level $k=p^\alpha$.
> The goal of this theorem and corollary was to derive an information theoretic lower bound in the SSL case. We will clarify this and the meaning of the two theoretical statements in the revision.
>
> Regarding Theorem 2.1, we agree that as originally stated it is weaker than the result of Donoho and Jin. The reason is that we considered the case where the probability of error in exact support recovery is bounded by 1/2. In the revision we will replace it by the following result (which still follows from the original proof without any required changes):
> Let $\delta \in (0,1)$. If $L < (1-\delta)\frac{2k}\lambda  \log(p-k+1)$, then for any support estimator $\hat S$ based on $\mathcal D_L$, $\Pr(\hat S\neq S) \geq \delta$.
> This results shows that exact support recovery is not possible with probability tending to one even for $\beta > 1-\alpha$. We will also discuss the relation between this result and those of Donoho and Jin (which are not about exact support recovery, but rather
> for approximate support recovery, namely $|\hat S \cap S|/k\to 1$ as $p\to \infty$).
>
> Regarding motivation for our conjecture: As derived by Donoho and Jin (05'), in the fully supervised (SL) setting there is a detection-recovery gap; Namely for a range of number of labeled samples, it is possible to detect that a sparse signal is present, but it is not possible to reliably recover its support. Intuitively, adding a few unlabeled samples should not resolve this gap. We will add a discussion about this issue in the revision.

---

### Official Review · Reviewer_NrxF · 2024-07-12

**Soundness:** 3
**Presentation:** 3
**Contribution:** 3
**Rating:** 6
**Confidence:** 3

**Summary:**

The paper studies semi-supervised learning in a simple mixture of two Gauss(ians setting.  Specifically, there is a uniform mixture of two Gaussians in $p$ dimensions with unknown means $N(\mu_1, I_p), N(\mu_{-1}, I_p)$.  We assume that the difference between the means $\Delta\mu = \mu_1 - \mu_{-1}$ is a $k$-sparse vector and for simplicity, we also assume all nonzero entries are $\pm \sqrt{\lambda/k}$ for some parameter $\lambda$ that controls the signal-to-noise ratio.  Now the learner receives both unlabeled samples from the mixture (which are just points in $\mathbb{R}^p$) and some labeled samples which also include the label of $+1$ or $-1$.  The goal of the learner is to learn the support of $\Delta \mu$ (say with at most $o(k)$ errors) , which then gives a classifier that separates the two Gaussians.  Information theoretically, with only labeled samples or only unlabeled samples, the optimal rates are well-understood (in a bit of an over-simplification and ignoring log factors, they are $\sim k/\lambda$ and $\sim k/\lambda^2$ respectively).  If we let $L$ and $n$ be the number of labeled and unlabeled samples, the paper shows that interpolating between the two rates, using a mixed number of labeled vs unlabeled samples cannot help.  Thus, up to constant factors, there is no benefit, information-theoretically, from combining labeled and unlabeled data.  This portion of the results are very similar to [Tifrea et. al. 2023], which proves the same type of result in the setting with no sparsity.

THe main contribution of the paper is identifying a regime where there are potential computational benefits to combining labeled and unlabeled data.  There is evidence (namely low-degree/SQ lower bounds) that solving the unsupervised variant of the problem is computationally hard with fewer than $k^2/\lambda^2$ samples.  Now consider a regime where we have a slightly subquadratic number, say $k^\gamma/\lambda^2$ for $\gamma < 2$, unlabeled samples. This paper shows that in this regime, even if we combine with a small amount of labeled data, the problem remains computationally hard.  On the other hand, the paper also shows that if we combine with a larger amount of labeled data (but still not enough to just solve the supervised setting by itself), then the problem becomes computationally tractable.  There is a constant factor gap between the amount of labeled data in the upper and lower bounds, but nevertheless, this identifies a region where combining labeled and unlabeled data makes the problem computationally tractable whereas using either only the labeled or unlabeled data individually would be intractable.

**Strengths:**

The paper makes advances in an important research direction of theory for understanding semi-supervised learning

The paper has a nice conceptual message in identifying a setting where there is a provable benefit to combining labeled and unlabeled data, proving both upper and lower bounds (under commonly believed computational assumptions).  It is also nice that the main algorithm gives a clean way to combine two different types of information.


The algorithm in the paper is simple enough to implement and the authors are able to support their conclusions with numerical experiments

**Weaknesses:**

The paper only studies a limited setting of a mixture of two Gaussians

The benefit from the unlabeled data, (in terms of reducing the number of labeled examples required), is only a fairly small (less than $2$) constant factor

It is a bit technical to actually describe/understand the setting where the algorithm in the paper has provable gains

**Questions:**

.

**Limitations:**

Yes

---

> ### Author Rebuttal · Authors · 2024-08-02
>
> We thank the reviewer for the thoughtful review.
>
> We agree that our work considers a limited case of two Gaussians. We remark that many previous authors also considered a same or similar setting. Indeed, it would be interesting to extend to larger number of Gaussians. However, as the SSL analysis was far from trivial already for only two Gaussians, such an extension is beyond the scope of this work.
>
> Regarding the benefit of unlabeled data. First, note that if $n=k^\gamma$ with $\gamma > 2$,
> recovering the support is computationally tractable using only the unlabeled samples.
> Indeed, in the regime $n=k^\gamma$ with $\gamma<2$ and $k=p^\alpha$, the unlabeled data decreases the number of labeled samples by a factor $(1-\alpha)/(1-\gamma\alpha)$. This factor can be much larger than 2 if $\alpha$ is close to 1/2 and $\gamma$ is close to 2. We will clarify this in the revision.
>
> The regime where our computationally efficient SSL provably works is the blue triangle in figure 1. We will add in the introduction a precise description of this region, in particular the formula
> for the blue line $\beta = 1-\gamma \alpha$.

---

> > ### Comment · Reviewer_NrxF · 2024-08-11
> >
> > Thank you for the response and addressing my concerns/questions.  Is it correct that if the parameters are bounded away from the boundaries of $\alpha = 1/2, \gamma = 2$ then the gain is only a constant factor?

---

> > > ### Author Response · Authors · 2024-08-11
> > >
> > > Yes, this is correct.

---

### Decision · Program_Chairs · 2024-09-25

**Decision:**

Accept (spotlight)

**Comment:**

Reviewers are unanimous to praise a theoretical contribution paper with an interesting and timely question in high-dimensional statistics/learning theory. It establishes the provable benefit in using jointly labeled and unlabeled data in the prototypical 2-class (sparse) Gaussian model in high dimension, which has generated a lot of attention in the recent years in order to establish precise phase diagrams for feasibility/infeasibility of learning in function of master parameters of the problem (dimension, sparsity, sample size -- here both for labeled and unlabeled data).
The reviews laud the high quality of writing of the paper (main paper presentation as well as clarity of proofs), the clear formulation of the overall message, the novelty and interesting technical aspects of the contribution, and the fact that the final picture is very thorough (upper and lower bounds)